# Regulation of NSL by TAF4A is critical for genome stability and quiescence of muscle stem cells

Angelina M. Georgieva [1], Krishna Sreenivasan[1], Dong Ding[1],
Clementine Villeneuve[2], Sara A. Wickström [2], Stefan Günther [1],
Carsten Kuenne [1], Ulrich Gärtner[3], Xinyue Guo[1], Yonggang Zhou[1],
Xuejun Yuan [1] ✉ & Thomas Braun [1,4] ✉

Acetylation of lamin A/C by the non-specific lethal complex, containing MOF and KANSL2, is instrumental for maintaining nuclear architecture and genome stability, but the mechanisms controlling expression of its components in different cell types are poorly characterized. Here, we show that TAF4A, primarily known as a subunit of TFIID, forms a complex with the heterotrimeric transcription factor NF-Y and is critical for cell type-specific regulation of *Kansl2* in muscle stem cells. Inactivation of *Taf4a* reduces expression of *Kansl2* and alters post-translational modification of lamin A/C, thereby decreasing nuclear stiffness, which disrupts the nuclear architecture and results in severe genomic instability. Reduced expression of *Kansl2* in *Taf4a*-mutant muscle stem cells changes expression of numerous genes involved in chromatin regulation. The subsequent loss of heterochromatin, in combination with pronounced genomic instability, activates muscle stem cells but impairs their proliferation, which depletes the stem cell pool and abolishes skeletal muscle regeneration. We conclude that TAF4A-NF-Y-dependent transcription regulation safeguards heterochromatin and genome stability of muscle stem cells via the non-specific lethal complex.

The nuclear lamina, situated below the inner nuclear membrane, is a proteinaceous network of intermediate filaments that forms widespread contacts with chromatin and proteins of the inner nuclear membrane[1]. It supplies mechanical stability to the nucleus and serves as an anchoring point for chromatin and transcription factors at the nuclear periphery, contributing to the organization of the eukaryotic genome into constitutive and facultative heterochromatin versus euchromatin[2]. Constitutive and facultative heterochromatin represent repressive states and are usually positioned at the nuclear periphery or attached to the nucleolus, whereas euchromatic is typically located in

the nuclear interior and contains actively transcribed genes. In addition to its role in transcriptional repression, the lamina-anchored heterochromatin provides mechanical stability to the nucleus[3]. Chromatin organization undergoes dynamic changes during activation, proliferation, and differentiation of stem cells. For example, skeletal muscle stem cells (MuSCs), also known as satellite cells, are characterized by a high content of facultative heterochromatin in the resting, non-activated state, which among other mechanisms is necessary to secure cellular quiescence[4]. Depletion of heterochromatin, e.g. by inactivation of the H4K20me2/3 methyltransferase

[1]Department of Cardiac Development and Remodeling, Max Planck Institute for Heart and Lung Research, 61231 Bad Nauheim, Germany. [2]Department of Cell and Tissue Dynamics, Max Planck Institute for Molecular Biomedicine, 48149 Münster, Germany. [3]Institute for Anatomy and Cell Biology, University of Giessen, 35392 Giessen, Germany. [4]Division of Life Science, Center for Stem Cell Research, The Hong Kong University of Science and Technology, Hong Kong, SAR, China. ✉e-mail: xuejun.yuan@mpi-bn.mpg.de; thomas.braun@mpi-bn.mpg.de

*Kmt5b*, prematurely activates MuSCs[5], thereby exhausting the stem cell pool with detrimental consequences, since MuSCs play an indispensable role for skeletal muscle growth and regeneration[5–7]. Furthermore, reduction of heterochromatin is a characteristic hallmark of aging[8].

Nuclear lamin genes, including *Lmna*, which encodes lamin A and lamin C (A-type lamins), as well as *Lmnb1* and *Lmnb2*, coding for lamin B1 and B2 (B-type lamins), provide the structural scaffold for the nuclear lamina. Mutations in the *Lmna* gene results in numerous different diseases, termed laminopathies, affecting striated muscles amongst other organs, or causing genomic instability and premature aging[9,10]. The function of lamins in chromatin organization and nuclear stability is not only compromised by mutations within the *Lmna* gene but also by defects in posttranslational modifications. Failure of lamin A/C acetylation due to dysfunction of the Non-Specific Lethal (NSL) chromatin-modifying complex, containing the lysine acetyltransferase MOF, results in increased solubility of lamins, defective phosphorylation, and accumulation of nuclear abnormalities[11]. Reduced nuclear stiffness and transient rupture of the nuclear envelope in *Lmna*-mutant MuSCs induces DNA damage, eventually causing cell death[12].

Relatively little is known about the regulation of epigenetic modifiers and components of the machinery that connect chromatin to the nuclear lamina and enable heterochromatin formation in the nuclear periphery of MuSCs. Accelerated displacement of nucleosomes in MuSCs due to reduced transcription of *Histone 2B* can be attributed to diminished expression of the transcription factor ATF3[7]; failure of facultative heterochromatin formation to inactivation of *Kmt5b*[13]. In both cases, genome instability, DNA damage, premature activation of MuSCs, and cellular senescence are the consequences. Increased DNA damage, either due to external cues or by deterioration of nuclear structures, will contribute to stem cell dysfunction during aging, not only in skeletal muscle but also in various organs, eventually compromising tissue regeneration[14]. The connection between morphological abnormalities of nuclei, accumulation of DNA damage, genome instability and compromised cellular functions is particularly evident in *LMNA* mutations, causing the Hutchinson-Gilford progeria syndrome[15]. A better understanding of critical upstream regulators preventing genome instability is mandatory to devise new strategies for avoiding the accumulation of DNA damage in stem cells.

Expression of components of the apparatus controlling functions of the nuclear lamina and heterochromatin formation in distinct cell types may not only be controlled by transcription factors but also by co-factors, which interact with core promoter recognition factors, such as TFIID. TFIID is part of the RNA polymerase II preinitiation complex and is composed of TBP and several subunits called TATA-binding protein Associated Factors (TBP-associated factors, or TAFs)[16]. Interestingly, TAFs also serve non-canonical functions in cell-type-specific transcription and gene regulation[17]. For example, the TAF12/TAF4 histone-fold heterodimer binds to the activation domain of MYB and protects MYB from degradation in cancer cells[18]. TAF4 can also provide docking sites for transcription factors and/or serve as a coactivator by transmitting signals from sequence-specific activators to other components of the basal transcription machinery. Inactivation of *Taf4a* inhibits differentiation of ES cells without affecting proliferation[19]. *Taf4a* is highly expressed in cortical neural stem cells (NSC) where it regulates homeostasis of NSCs by interacting with RanBPM[20]. In MuSCs, the function of *Taf4a* remains elusive.

Here, we searched for factors controlling the expression of chromatin modifiers and components of the nuclear lamina, facilitating heterochromatin formation in MuSCs. We found that *Taf4a* is highly expressed in MuSCs but to a lesser extent in differentiated myofibers. Inactivation of *Taf4a* prematurely activates adult MuSCs and depletes the MuSC pool, impairing skeletal muscle regeneration. We also discovered that TAF4A forms a complex with the transcription factor NF-Y, which is crucial for the expression of *Kansl2*, an essential subunit of the NSL complex that stabilizes the nuclear architecture by acetylation of lamin A/C. Absence of *Taf4a* increases DNA damage and nuclear abnormalities, indicated by nuclear blebbing and formation of micronuclei, favoring genome instability, thereby decreasing proliferation and increasing apoptosis.

## Results

### Inactivation of *Taf4a* depletes heterochromatin in adult MuSCs and induces premature activation

We previously discovered that loss of heterochromatin, induced by inactivation of *Kmt5b*, promotes exit of MuSCs from quiescence[5]. Loss of heterochromatin in *Kmt5b*-deficient MuSCs leads to aberrant transcription during S-phase, facilitating transcription replication collisions and genome instability[7]. We reasoned that transcriptional regulation of chromatin modifiers controlling heterochromatin formation is decisive to keep MuSCs in a quiescent state and to maintain nuclear architecture. To identify transcription factors or transcriptional co-factors that may play a role in the regulation of chromatin modifiers in different stem cell populations, including MuSCs, we examined the transcriptional landscape of different cell populations. Analysis of the RNA-Seq CAGE (Cap Analysis of Gene Expression) database in RIKEN FANTOM5 project[21,22] revealed high expression of the transcriptional co-factor *Taf4a* particularly in stem cells and to a lesser extent in neurons, Schwann cells, macrophages, and megakaryocytes (Supplementary Fig. 1a). To analyze expression of *Taf4a* in MuSCs, we interrogated our RNA-seq datasets of MuSC and muscle tissue[23] and found substantially higher levels of *Taf4a* mRNA in freshly isolated MuSCs (fiMuSCs) compared to whole muscles, which mainly contain muscle fibers (Supplementary Fig. 1b). Immunofluorescence staining for TAF4A using FACS-isolated MuSCs from Pax7[zsgreen] reporter mice and MuSCs situated on freshly isolated single myofibers revealed high levels of TAF4A in quiescent and proliferating MuSCs. In contrast, levels of TAF4A in nuclei of differentiated myofibers were substantially lower (Supplementary Fig. 1c).

To investigate the role of *Taf4a* in MuSCs, we generated MuSC-specific conditional *Taf4a* knockout mice by crossing Pax7CRE[ERT] to Taf4a[flox/flox] mice, yielding Pax7CRE[ERT]/Taf4a[flox/flox] animals (hereafter referred to as *Taf4a[sKO]*). Isolated *Taf4a[sKO]* MuSCs show an efficient deletion of exons 11-12 after administration of tamoxifen (TAM) (Supplementary Fig. 2a), resulting in a virtually complete absence of *Taf4a* transcripts and protein (Supplementary Fig. 2b, c). *Taf4a[sKO]* mice were viable with no apparent changes in body and skeletal muscle weights. No obvious morphological changes of skeletal muscle tissue were detected in H&E-stained sections of the tibialis anterior muscle (TA) under baseline conditions (Supplementary Fig. 2d–f). However, electron microscopy (EM) analysis revealed a strong reduction of the heterochromatin content in *Taf4a[sKO]* MuSCs, which is a hallmark of loss of MuSC quiescence (Fig. 1a). Reduction of the heterochromatin content was further assessed by immunofluorescence staining with H3K9me3 antibodies, demonstrating a dramatic decrease of H3K9me3 levels in chromocenters and the nuclear periphery (Fig. 1b). Likewise, western blot analysis showed a substantial reduction of the heterochromatin mark H3K9me3 and to a lesser extent of H4K20me3 and H3K27me3 in *Taf4a[sKO]* MuSCs (Fig. 1c). In line with the decrease of heterochromatin, we detected elevated levels of *MyoD* mRNA, a marker for MuSC activation, in freshly isolated in *Taf4a[sKO]* MuSCs (Fig. 1d) and the number of PAX7/MYOD double-positive cells was increased in *Taf4a[sKO]* muscles (Fig. 1e). Analysis of calcitonin receptor (CalcR) expression, a marker for quiescent MuSCs, revealed a decline of the number of CalcR/PAX7 double-positive cells, supporting the conclusion that *Taf4a* is required for maintaining quiescence of MuSCs (Fig. 1f). To obtain a better understanding of the underlying transcriptional changes, we compared gene expression profiles of freshly isolated MuSCs from control and *Taf4a* mutant mice by RNA-seq.

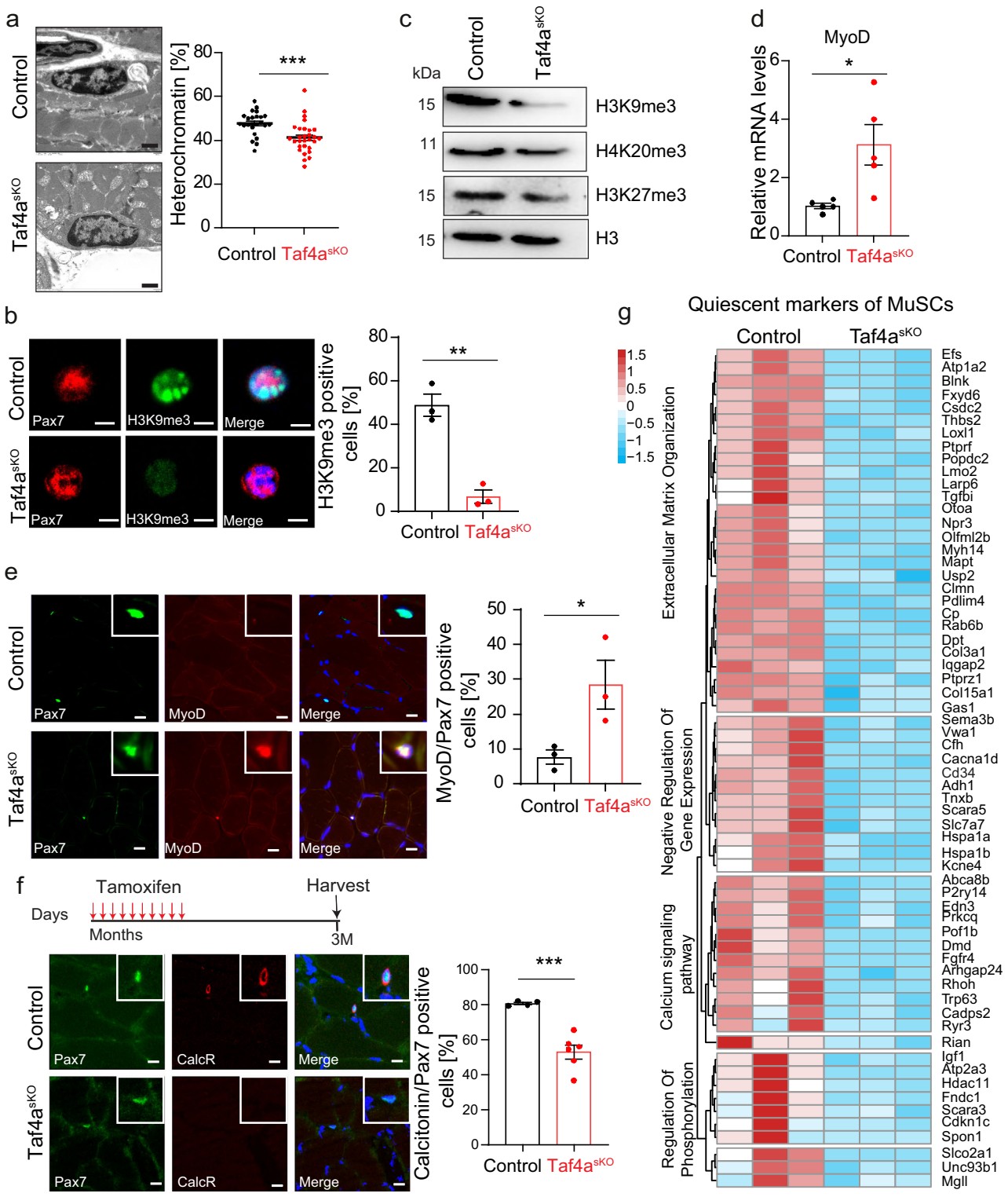

Bioinformatics analysis identified 1364 downregulated and 1173 upregulated genes in *Taf4a^sKO* compared to control MuSCs (Supplementary Fig. 3a). Numerous genes, characteristic for quiescent MuSCs were downregulated after inactivation of *Taf4a*[24], including genes related to extracellular matrix organization, negative regulation of gene expression, calcium signaling, and regulation of phosphorylation (Fig. 1g), confirming MuSC activation after inactivation of *Taf4a*. The majority of DEGs was related to premature activation of MuSCs, but GO analysis of the 2537 differentially expressed genes (DEGs) also identified several additional terms, comprising downregulated genes related to

ribosome, cell cycle and DNA replication, which are normally upregulated upon physiological activation of MuSCs (Supplementary Fig. 3b). Furthermore, genes regulated by the PI3K-Akt pathway such as *Cdkna1(p21)* and *Bcl2l1*, which control cell cycle arrest, showed higher expression in *Taf4a^sKO* MuSCs (Supplementary Fig. 3c), suggesting that absence of *Taf4a* not only abrogates quiescence but also causes additional changes.

To determine whether the reduction of heterochromatin in *Taf4a^sKO* MuSCs is directly associated with modified gene expression and occurs at specific genomic regions, we performed ATAC-seq of

**Fig. 1 | *Taf4a^sKO* MuSCs are characterized by depletion of heterochromatin and loss of quiescence. a** EM images showing reduced heterochromatin content in MuSCs of *Taf4a^sKO* mice. Each dot represents a single MuSC (*n* = 24 MuSCs in control; *n* = 29 MuSCs in *Taf4a^sKO* mice). Scale bar: 2 μm. Quantifications are shown in the right panel (unpaired two-tailed *t* test: \*\*\**p* = 0.0006, *n* = 3). **b** Immunofluorescence staining of H3K9me3 in freshly isolated MuSCs from control and *Taf4a^sKO* mice. Scale bar represents 5 μm. Percentage of H3K9me3/Pax7 positive MuSCs are shown in the right panel (unpaired two-tailed *t* test: \**p* = 0.0020, *n* = 3). **c** Western blot analysis of histone modifications H3K9me3, H4K20me3, and H3K27me3 in freshly isolated MuSCs from control and *Taf4a^sKO* mice (*n* = 2). H3 was used as loading control. **d** RT-qPCR of *MyoD* expression level in control and *Taf4a^sKO* freshly isolated MuSCs (*n* = 5). *m36b4* was used as a reference gene (unpaired two-tailed *t* test: \**p* = 0.0161, *n* = 5). **e** Immunofluorescence staining of MYOD/PAX7

double-positive MuSCs in cryo-sections of control (*n* = 3) and *Taf4a*-deficient (*n* = 3) TA muscles. Relative quantification of MYOD-positive cells in 50 PAX7+ cells per mouse is shown in the right panel. (unpaired two-tailed t-test: \**p* = 0.0476). **f** Immunofluorescence staining of CalcR + / PAX7 double-positive MuSCs in control (*n* = 4) and *Taf4a*-deficient (*n* = 6) TA muscles. Relative quantification of CalcR+ cells in 50 PAX7+ cells is shown in the right panel (unpaired two-tailed *t* test: \*\*\**p* = 0.0006). **g** Heat map of MuSC quiescence genes in control and *Taf4a^sKO* MuSCs (*n* = 3). 8–20-week-old males and females were used. Muscles and/or MuSCs in (**a**–**e**) were isolated 2–6 weeks after tamoxifen administration. TA muscles were isolated 3 months after tamoxifen administration in (**f**). Data are presented as mean ± SEM of biological replicates. Source data are provided in the Source Data file.

freshly isolated MuSCs from control and *Taf4a^sKO* mice. The coverage heat map indicated a more open chromatin organization in *Taf4a^sKO* MuSCs (Supplementary Fig. 3d). Bioinformatics analysis identified 5330 gained and 4068 lost ATAC-seq peaks in *Taf4a^sKO* compared to control MuSCs. However, only 13–15% of altered peaks were annotated to genes, suggesting that changes primarily occur in repetitive and intergenic regions and to a lesser degree in regions that are directly relevant for regulation of gene expression (Supplementary Fig. 3e). Further analysis of the genomic distribution of peaks showed that only 3% are located in proximal promoter regions, whereas 51% are within intronic regions and 38% within intergenic regions (Supplementary Fig. 3f). Importantly, the correlation between transcriptional changes and differential ATAC-seq peaks was comparatively weak. Only 6.5% (165 out of 2371 DEGs) of transcriptionally altered genes were associated with changes in DNA accessibility, suggesting that the altered chromatin accessibility in *Taf4a^sKO* MuSCs most likely is caused by indirect effects (Supplementary Fig. 3g, h). Taken together, the comparative ATAC-seq analysis indicated that changes in chromatin accessibility after inactivation of *Taf4a* primarily happen in intronic and intergenic regions, most likely due to untimely activation of MuSCs and loss of H3K9me3. A strong link to changes in gene expression was not apparent.

## Deregulation of epigenetic modifiers in *Taf4a^sKO* MuSCs mainly occurs independent of premature MuSC activation

The substantial loss of heterochromatin in *Taf4a^sKO* MuSCs could be caused by direct regulation of epigenetic modifiers by TAF4A, indirect effects due to premature activation of MuSCs resulting in heterochromatin reduction, or both. To address these possibilities, we compared differentially expressed genes in freshly isolated *Taf4a^sKO* MuSCs and activated, proliferating WT MuSCs to freshly isolated WT MuSCs. We detected 269 upregulated and 484 downregulated genes in both *Taf4a^sKO* and activated, proliferating WT MuSCs, suggesting that differential expression of these genes might be an indirect effect due to premature activation of MuSCs (Fig. 2a). 428 genes were downregulated in *Taf4a^sKO* MuSCs but upregulated in activated WT MuSCs (Fig. 2a). Gene Ontology (GO) enrichment analysis of these 428 genes revealed association with G1 to S cell cycle control, mechanisms associated with pluripotency, p53 signaling, and delta notch signaling pathway (Fig. 2b). Comparison of epigenetic modifiers uncovered 21 genes with lower expression in freshly isolated *Taf4a^sKO* MuSCs relative to freshly isolated WT MuSCs, which makes these genes potential direct targets of TAF4A (Fig. 2a, epigenetic modifiers in brackets). Interestingly, two genes of this group of 21, *Mcrs1* and *Kansl2*, code for members of the chromatin-modifying NSL complex (Fig. 2c, labeled in green), albeit we also observed reduced expression of epigenetic modifiers involved in heterochromatin formation, such as *Smyd5* and *Suv39h2* (Fig. 2c–e). Accordingly, we observed a substantial reduction of the SUV39H2-dependent heterochromatin mark H3K9me3 in activated, proliferating *Taf4a^sKO* MuSCs compared to activated, proliferating WT MuSCs by immunofluorescence and western blot analysis

(Fig. 2f, g). Taken together, inactivation of *Taf4a* in MuSCs severely disrupts expression of genes for epigenetic modifiers, independent of the activation of MuSCs.

## TAF4A forms a complex with NF-Y to regulate *Kansl2*, a key component of the NSL complex

To distinguish between direct targets, whose expression depends on the binding of TAF4A to their regulatory regions, and genes with differential expression due to indirect effects, we performed CUT&RUN experiments. We identified 928 gene-annotated peaks in wild-type MuSCs (Fig. 3a), of which approximately 58% localize within proximal promoter regions and 18% within intron regions (Fig. 3b). Previous studies suggested that TAF4A preferentially promotes transcription from downstream core promoter element (DPE)-containing promoters without a TATA-box. According to these studies, TATA-containing, DPE-less promoters are far less dependent on TAF4A[25]. Surprisingly, our analysis revealed that TAF4A-peaks are present at all well-known core promoter elements, without any obvious preferences (Fig. 3c). Closer examination of TAF4A CUT&RUN peaks revealed a strong overlap with SP1 and NF-YA transcription factor binding site motifs (Fig. 3d). Protein-protein interactions have been described before between SP1 and TAF4A, which may explain the enrichment of TAF4A at SP1-binding sites[26]. In contrast, interactions between TAF4A and NF-YA have not been reported yet, although cooperative binding of SP1 and NF-Y to adjacent motifs and interactions between SP1 and NF-YA have been documented before[27].

NF-Y is a ubiquitously expressed heterotrimeric transcription factor (TF) complex, containing NF-YA, NF-YB, and NF-YC subunits. NF-YA has DNA binding and transactivation domains, whereas NF-YB and NF-YC carry histone-fold domains (Supplementary Fig. 4a). We found that *Nf-ya* is predominately expressed in freshly isolated MuSCs but to a lower extent in muscle tissue, similar to *Taf4a* (Fig. 3e), consistent with a previous study demonstrating absence of NF-YA protein in the nuclei of skeletal myotubes and cardiomyocytes[28]. Co-immunoprecipitation experiments revealed interactions of TAF4A with NF-YA and NF-YB in C2C12 cells and primary MuSCs, demonstrating that TAF4A associates with the NF-Y transcription factor complex (Fig. 3f, Supplementary Fig. 4b). The NF-Y complex probably recruits TAF4A to specific genomic sites via its DNA-binding domain, most likely independent of TFIID. To further investigate this possibility, we performed CUT&RUN assays in WT and *Taf4a^sKO* MuSCs using an NF-YA antibody. NF-YA peaks were detected at 5146 genes, substantially more than for TAF4A in WT MuSCs (928 peaks) (Supplementary Fig. 4c). 34% of NF-YA binding occurred at promoters and 34% at intronic regions, which is in line with published findings, indicating that NF-YA localizes at both promoter and enhancer elements (Supplementary Fig. 4d, e). 77.35% of all TAF4A CUT&RUN peaks coincided with NF-YA peaks and 62.87% were found at SP1 binding motifs (Fig. 3g). 159 genes bound by both TAF4A and NF-YA were differentially expressed in *Taf4a^sKO* MuSCs, suggesting that TAF4A and NF-Y collaborate in the regulation of these genes (Fig. 3h).

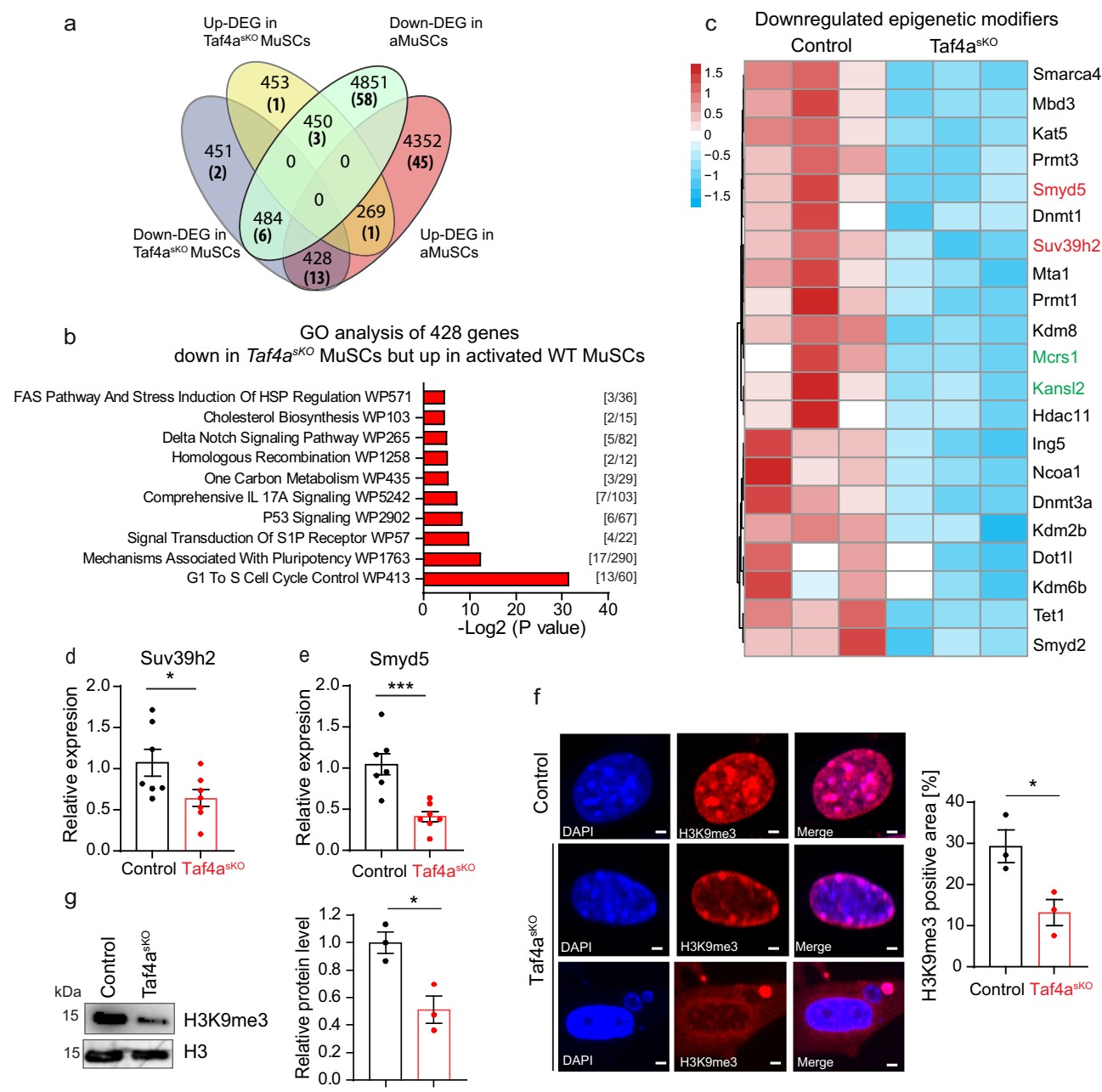

**Fig. 2 | Inactivation of *Taf4a* in MuSCs causes a broad deregulation of epigenetic modifiers. a** Venn diagram of up- and down-regulated genes in freshly isolated *Taf4a^sKO* MuSCs and WT activated MuSCs. Number of deregulated epigenetic modifiers are shown in brackets. **b** GO term analysis of 428 genes, showing an overlap between downregulated genes in *Taf4a^sKO* MuSC and upregulated genes in WT activated MuSCs based on *P*-values using EnrichR (Fisher's exact test, two-sided). **c** Heat map of 21 epigenetic modifiers, which might be directly regulated by TAF4A in *Taf4a^sKO* MuSCs (*n* = 3). **d**, **e** RT-qPCR analysis of *Suv39h2* (**d**) and *Smyd5* (**e**) expression in freshly isolated control and *Taf4a^sKO* MuSCs. m36b4 was used as a reference gene (unpaired two-tailed *t* test, (**d**) \**p* = 0.0487; (**e**) \*\*\**p* = 0.007, *n* = 7). **f** Immunofluorescence staining of H3K9me3 in cultured MuSCs from control and *Taf4a^sKO* mice. Scale bar: 2 μm. Quantification of H3K9me3 nuclear areas is shown in the right panel (unpaired two-tailed *t* test: \**p* = 0.0323, *n* = 3). **g** Western blot analysis of H3K9me3 in cultured MuSCs from control and *Taf4a^sKO* mice. H3 was used as a loading control (unpaired two-tailed *t* test: \**p* = 0.0172, *n* = 3). 8–20-week-old males and females were used. MuSCs were isolated 4–6 weeks after tamoxifen administration. Data are presented as mean ± SEM of biological replicates. Source data are provided in the Source Data file.

Inactivation of *Taf4a* in MuSCs did not substantially affect the binding of NF-YA to regulatory regions. Only 48 NF-YA peaks were lost and 71 peaks were gained in *Taf4a^sKO* MuSCs compared to WT MuSCs, indicating that NF-YA detects and binds cognate sites independent of TAF4A (Supplementary Fig. 4f). For example, deletion of *Taf4a* in MuSCs did not change binding of NF-YA to the *Kansl2* promoter (Supplementary Fig. 4g). Further GO analysis of the 159 DEGs bound by TAF4A and NF-YA revealed potential involvement of these DEGs in translation, gene expression, apoptotic processes, and positive regulation of transcription (Supplementary Fig. 4h). The majority of differentially expressed chromatin modifiers in *Taf4a^sKO* MuSCs was neither bound by TAF4A nor by NF-Y, suggesting that differential expression of these genes was due to secondary effects of *Taf4a* inactivation, most likely the loss of MuSC quiescence. However, the differentially expressed chromatin modifiers *Kansl2*, *Mbd3*, and *Prmt1* appear to be direct targets of TAF4A, since they were bound by both

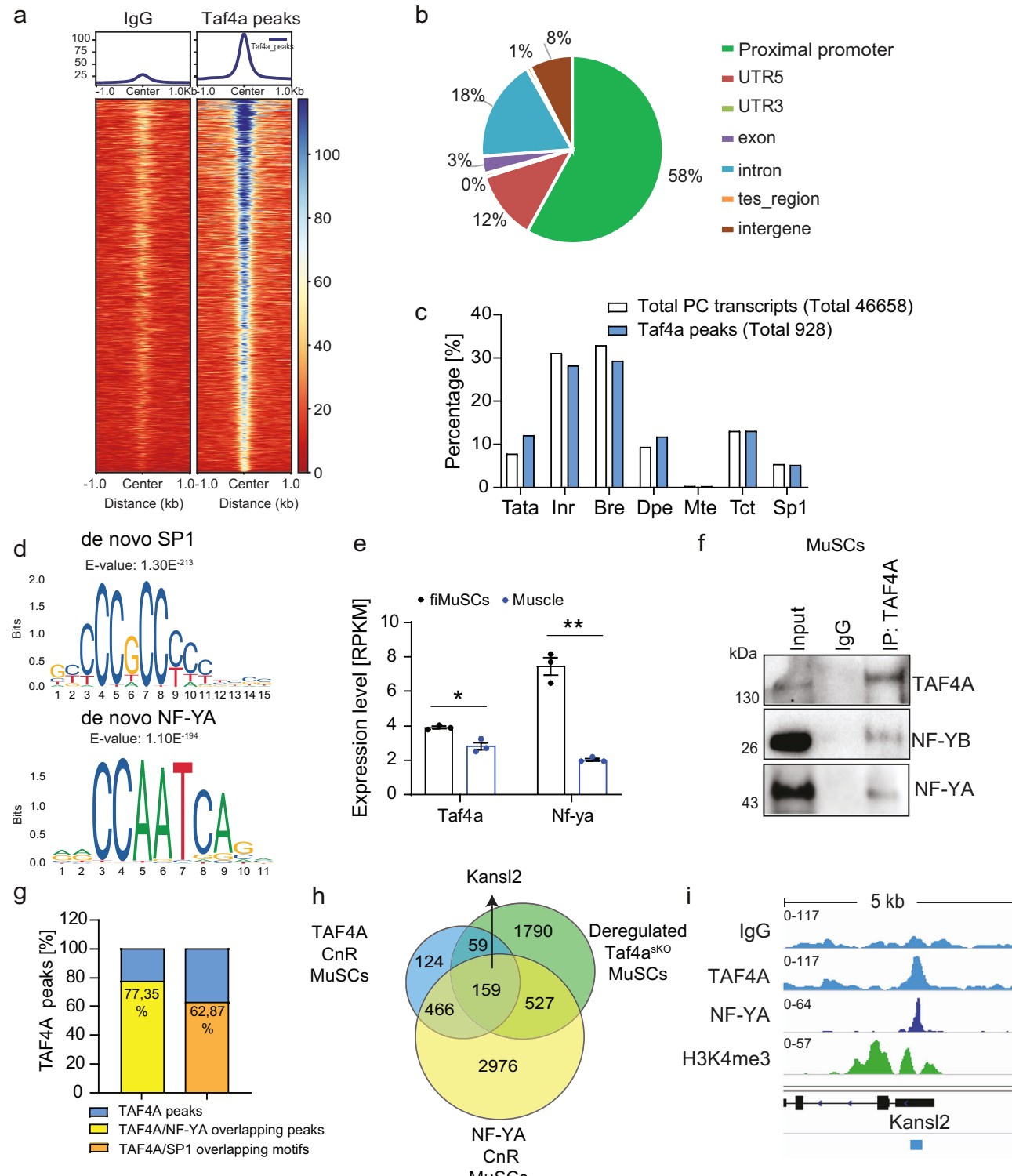

**Fig. 3 | TAF4A interacts with NF-Y and binds to the promoter of *Kansl2*. a** Heat maps of centered TAF4A CUT&RUN peaks. The blue-to-red gradient indicates high-to-low counts in the corresponding region. **b** Genome-wide distribution of TAF4A peaks. **c** Bar plot representing TAF4A peaks at different types of promoters. **d** Detected SP1 and NF-YA binding site under TAF4A CUT&RUN peaks. **e** *Taf4a* and *Nf-Ya* expression in freshly isolated MuSCs (fiMuSCs) and muscle based on reads per exon kilobase per million (RPKM) values (unpaired two-tailed *t* test: \**p* = 0.0103, \*\*\**p* = 0.0005 *n* = 3). **f** Co-IP of TAF4A and NF-YA/B in wildtype MuSCs. Blots were probed with antibodies against TAF4A and NF-YA or NF-YB (*n* = 2). **g** Percentage of TAF4A CUT&RUN peaks overlapping either with NF-YA peaks or with SP1 binding motifs. **h** Venn diagram of overlapping TAF4A CUT&RUN (CnR) peaks, NF-YA CUT&RUN (CnR) peaks, and RNA-seq DEG in *Taf4a^sKO* MuSCs. **i** TAF4A, NF-YA and H3K4me3 distribution in the proximal promoter region of the *Kansl2* gene in WT MuSCs. 8-20-weeks-old males and females were used. Data are presented as mean ± SEM of biological replicates. Source data are provided in the Source Data file.

TAF4A and NF-Y and showed downregulation in MuSCs after inactivation of *Taf4a*. Among the differentially expressed chromatin modifiers *Kansl2* caught our attention. *Kansl2* is an essential component of Non-Specific Lethal (NSL) chromatin-modifying complex, which plays important roles in the regulation of transcription and organization of chromatin in the nuclear periphery[11]. We localized overlapping TAF4A and NF-YA peaks at the transcriptional initiation site (TSS) of the *Kansl2* gene flanked by histone H3K4me3, which together with the reduced expression of *Kansl2* after inactivation of *Taf4a* indicates that the TAF4A-NF-Y complex directly regulates *Kansl2* expression (Fig. 3i).

## KANSL2 controls genes involved in chromatin organization of MuSCs

KANSL2 is an integral subunit of the NSL chromatin-modifying complex containing the histone acetyltransferase MOF, which catalyzes acetylation of H4K16 for the regulation of transcriptional initiation. MOF also mediates acetylation of lamin A/C, required for stability and maintenance of the nuclear architecture[11,29] (Fig. 4a). Targeted comparison of expression levels for NSL components in freshly isolated *Taf4a^sKO* and WT MuSCs uncovered that loss of *Taf4a* primarily affected *Kansl2* and *Mcrs1*. Other NSL components showed no significant downregulation or were even upregulated, such as *Ogt* (Fig. 4b). The reduced expression of *Kansl2* and *Mcrs1* in *Taf4a^sKO* MuSCs was confirmed by RT-qPCR and western blot analysis (Fig. 4c, d, Supplementary Fig. 5a). In contrast to the *Kansl2* promotor (Fig. 3i), TAF4A and NF-YA did not bind to the *Mcrs1* promoter in WT MuSC (Supplementary Fig. 5b). Instead, we observed a prominent peak for KANSL2 (Supplementary Fig. 5b), suggesting that the downregulation of *Mcrs1* in *Taf4a^sKO* MuSCs is an indirect event, not directly dependent on the absence of TAF4A but on the reduced expression of *Kansl2* and/or other secondary effects.

Next, we wanted to explore whether the downregulation of *Kansl2* contributes to the loss of heterochromatin, loss of quiescence, and transcriptional deregulation in *Taf4a^sKO* MuSCs. We reasoned that these profound changes might be in large part caused by the disrupted function of the NSL complex. In fact, we detected reduced acetylation of H4K16, a target of NSL, in *Taf4a^sKO* MuSCs, confirming compromised function of the NSL complex, secondary to the loss of TAF4A in MuSCs (Fig. 4e). CUT&RUN experiments using WT MuSCs identified more than 7500 KANSL2-bound genes (Fig. 4f). 44% of KANSL2 peaks localized in introns, 31% in intergenic regions, and 16% at promoter regions (Fig. 4g). Importantly, approximately 45% of DEGs in *Taf4a^sKO* MuSCs showed prominent KANSL2 peaks (Fig. 4h), confirming the important role of KANSL2 in TAF4-dependent transcriptional changes. The reduced expression of *Kansl2* in absence of TAF4A does also seem to account for the deregulation of numerous epigenetics modifiers in *Taf4a^sKO* MuSCs as indicated by the presence of KANSL2 peaks on the corresponding genes in WT MuSCs (Fig. 4i). The same holds true for the regulation of the NSL-component MCRS1, whose promoter carries a prominent KANSL2 peak (Supplementary Fig. 5b). Other KANSL2-bound downregulated genes are involved in cell cycle control and DNA replication as indicated by GO-term analysis, whereas KANSL2-bound upregulated genes are involved in the p53 pathway and apoptosis (Fig. 4j).

## The TAF4A-NFY-NSL axis safeguards genomic stability of MuSCs

In addition to its role as chromatin modifier for regulating transcription, the NSL complex controls stability of the nuclear architecture by acetylating lamin A/C (Fig. 4a). Loss of lamin A/C acetylation leads to impaired nuclear mechanostability, resulting in nuclear abnormalities and catastrophic genomic instability, called chromotripsis[11]. We observed a large number of nuclear abnormalities in *Taf4a^sKO* MuSCs, including nuclear blebbing and formation of micronuclei, indicating catastrophic genomic instability and compromised nuclear

mechanical integrity (Fig. 5a). We also found increased numbers of γH2A.X+ MuSCs and elevated γH2A.X protein levels in *Taf4a^sKO* MuSCs (Fig. 5b, c), suggesting accumulation of DNA damage. Importantly, the level of Lamin A/C acetylation was reduced and the level of phospho-Ser 392 lamin A/C was strongly increased in *Taf4a^sKO* MuSCs, which is indicative of increased solubility of lamin A/C (Fig. 5d, e). We also observed a similar reduction of lamin acetylation in *Kansl2* knockdown cells, confirming the role of TAF4A and KANSL2 in regulating lamin A/C acetylation (Supplementary Fig. 5c). Since increased solubility of lamin A/C has a direct impact on the stiffness of nuclei[11], we assessed the elasticity of nuclei in MuSCs by atomic force microscopy (AFM)-based force indentation spectroscopy. Computation of the effective elastic modulus of nuclei uncovered a significant reduction of the stiffness of nuclei in *Taf4a^sKO* MuSCs compared to controls (Fig. 5f). To further prove the key role of the TAF4A-NFY complex for regulation of NSL, we knocked down *Nf-ya* in WT MuSCs using shRNAs (Supplementary Fig. 5d). Knockdown of *Nf-ya* in MuSCs reduced expression of *Kansl2*, confirming that NF-YA collaborates with TAF4A to transcriptionally activate *Kansl2* (Fig. 5g). Similar results were obtained when analyzing publicly available RNAseq datasets for *Nf-ya* knockout MuSCs (Supplementary Fig. 5e). As expected, knockdown of *Nf-ya* reduced expression of *Mcrs1*, which was reversed by overexpression of *Kansl2* (Supplementary Fig. 5f). Importantly, knockdown of *Nf-ya*, *Kansl2*, or *Mcrs1* (Supplementary Fig. 5d, g, h) increased nuclear abnormalities of MuSCs, corroborating the instrumental role of the *NSL* complex for preventing genomic instability in *Taf4a^sKO* MuSCs (Fig. 5h), whereas overexpression of *Kansl2* strongly reduced formation of micronuclei and normalized nuclear stiffness in *Taf4a^sKO* MuSCs (Fig. 5i, j). Consistently, overexpression of *Kansl2* after knockdown of *Nf-ya* KD MuSCs also prevented micronuclei formation (Fig. 5k, l). Taken together, the results demonstrate that TAF4A, in combination with NF-YA, drives expression of *Kansl2*, a critical component of the NSL complex, which is instrumental in maintaining structural integrity of the nuclear architecture by acetylating lamin A/C.

## Inactivation of *Taf4a* abrogates muscle regeneration

We assumed that premature activation of MuSCs in *Taf4a^sKO* MuSCs, in combination with aberrant gene regulation, compromised mechanics of nuclei, and genomic instability, impairs maintenance and proliferation of MuSCs, eventually preventing successful skeletal muscle regeneration. To investigate whether *Taf4a* is required for MuSC expansion, single isolated myofibers with attached MuSCs from control and *Taf4a^sKO* mice were cultured and pulse-chase labeled with EdU for three hours. We observed a dramatic reduction of EdU + /PAX7+ cells and the number of MuSC colonies on cultured myofibers from *Taf4a*-deficient MuSCs dropped strongly (Fig. 6a, b). Similarly, EdU incorporation of isolated MuSCs from *Taf4a^sKO* mice was markedly reduced, indicating decreased proliferation (Fig. 6c). Likewise, knockdown of *Taf4a* in MuSCs decreased EdU incorporation (Supplementary Fig. 6a, b). In contrast, knockdown of *Taf4a* in mouse embryonic stem cells (mESCs) (Supplementary Fig. 6c) or in mouse embryonic fibroblasts (MEFs) (Supplementary Fig. 6d) did not decrease proliferation, indicating cell-specific functions of *Taf4a*. Unlike the experiment with MEFs, assessment of mESC proliferation was only repeated twice. Thus, no statistical evaluation was done. The normal proliferation of MEFs after knockdown of *Taf4a*, which even increased in some experiments compared to control MEFs (Supplementary Fig. 6d), corresponded to normal levels of *Kansl2* expression (Supplementary Fig. 6e). Furthermore, the number of MEFs with micronuclei did not increase after knockdown of *Taf4a* (Supplementary Fig. 6f). Comparison of transcriptional changes using publicly available RNAseq datasets from MuSCs, MEFs, keratinocytes, and liver cells showed a relatively low overlap of DEGs among different cell types (Supplementary Fig. 6g), confirming cell type-specific activity of TAF4A.

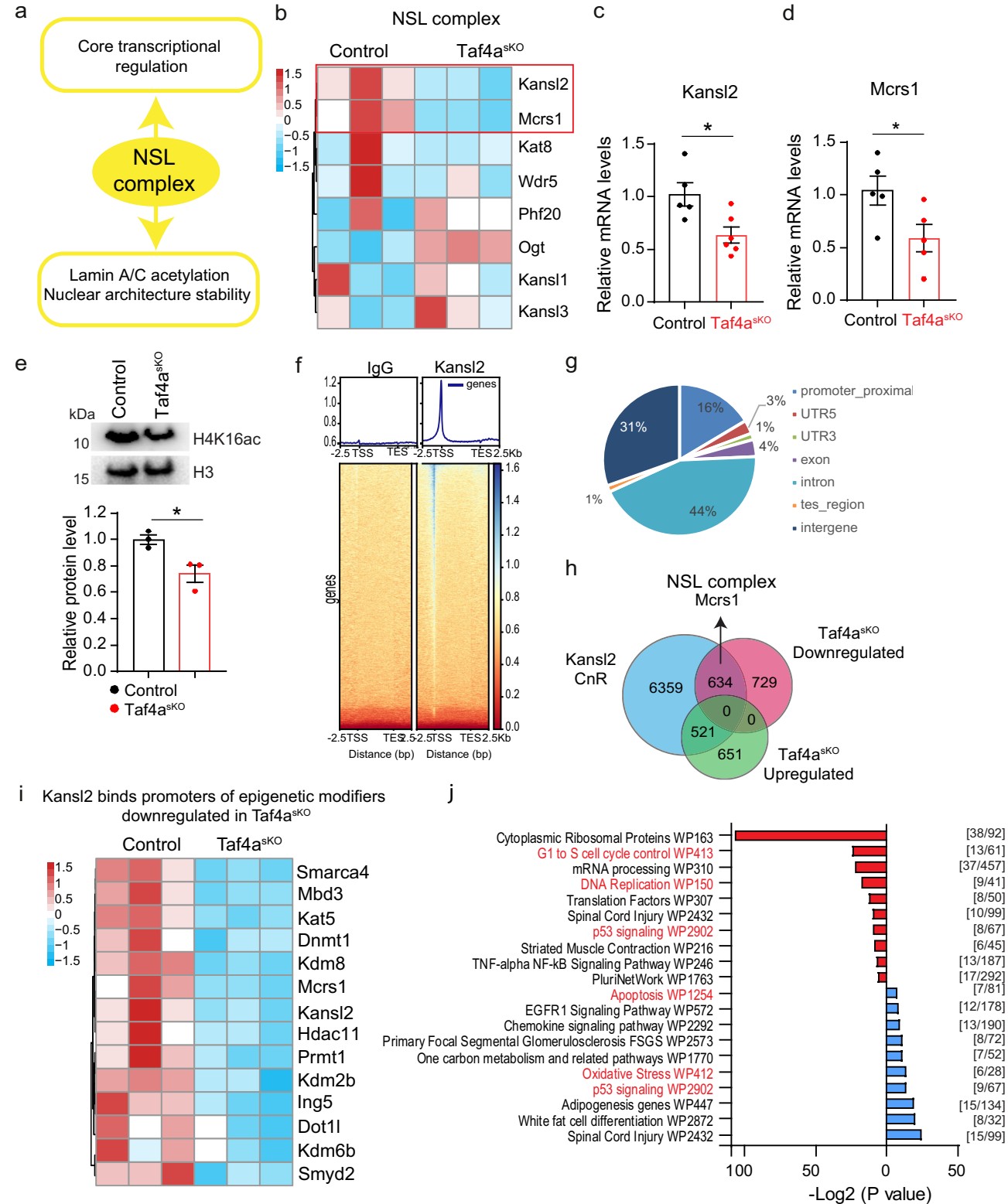

i Kansl2 binds promoters of epigenetic modifiers downregulated in Taf4a^sKO

Loss of heterochromatin resulting in loss of MuSCs quiescence normally reduces the muscle stem cell pool over time[5]. Since one of the most prominent features of *Taf4a^sKO* MuSCs is the combined loss of heterochromatin and cellular quiescence but also reduced proliferation, we studied the potential depletion of the MuSC pool in *Taf4a^sKO* mice. We observed a substantial reduction of PAX7[+] MuSCs in TA muscles after *Taf4a* inactivation (Fig. 6d). The reduction of MuSCs was accompanied by an increase of apoptotic MuSCs in culture as

measured by the TUNEL assay (Fig. 6e), and increased p53 protein levels (Fig. 6f). We concluded that the depletion of MuSCs in *Taf4a^sKO* mice is most likely the combined effect of the loss of heterochromatin and quiescence, which compromises MuSC self-renewal and the dramatic increases genomic instability, causing cell death. Finally, we subjected *Taf4a^sKO* and control mice to acute muscle injury by injection of CTX into the TA muscle (Fig. 6g). The weight ratio of damaged compared to undamaged TA muscles weight was strongly reduced in

**Fig. 4 | Taf4a regulates expression of Kansl2. a** Representation of NSL complex functions. **b** Heat map of expression levels of NSL complex components in *Taf4a^sKO* MuSCs compared to control MuSCs (*n* = 3). **c, d** RT-qPCR analysis of *Kansl2* (**c**) and *Mcrs1* (**d**) expression in freshly isolated control and *Taf4a^sKO* MuSCs. m36b4 was used as reference gene (unpaired two-tailed *t* test, (**c**) *p = 0.0161, (**d**) *p = 0.0417, *n* = 5). **e** Western blot analysis of H4K16ac in cultured MuSCs from control and *Taf4a^sKO* mice. H3 was used as loading control (unpaired two-tailed *t* test: *p = 0.0280, *n* = 3). **f** Heat maps of CUT&RUN signals of KANSL2 at transcriptional start sites. The blue-to-red gradient indicates high-to-low counts in the corresponding region. **g** Genome-wide distribution of KANSL2 peaks. **h** Venn diagram of overlapping KANSL2 CUT&RUN peaks, up and down-regulated genes from RNA-seq data in *Taf4a^sKO* MuSCs. **i** Heat map showing KANSL2 binding at promoters of downregulated epigenetic modifiers genes in *Taf4a^sKO* MuSCs (*n* = 3). **j** GO term analysis of genes with an overlap between KANSL2 CUT&RUN peaks and down-regulated genes in *Taf4a^sKO* MuSCs based on P-values using EnrichR (Fisher's exact test, two-sided). The number of genes associated with each GO term is indicated in square brackets. 8–20-week-old males and females were used. MuSCs were isolated 4–6 weeks after tamoxifen administration. Data are presented as mean ± SEM of biological replicates. Source data are provided in the Source Data file.

*Taf4a^sKO* mice 14 days after injury, indicating impaired muscle regeneration whereas control mice completely restored the muscle weight (Fig. 6g). Hematoxylin and eosin staining revealed a strong reduction of newly formed myofibers in *Taf4a^sKO* mice 14 days after CTX injection, while muscle tissue architecture was completely restored in control mice (Fig. 6h). Instead of newly generated muscle fibers with centralized nuclei, Masson's Trichrome demonstrated a massive increase of fibrosis in damaged *Taf4a^sKO* muscles, replacing myofibers (Fig. 6h), revealing an indispensable function of *Taf4a* for skeletal muscle regeneration.

## Discussion

The ability to maintain quiescence and genomic stability is an essential feature of healthy stem cells. Loss of quiescence but also reduced genomic stability depletes the MuSC pool and eventually incapacitates skeletal muscle regeneration. Loss of quiescence and genomic instability are both connected to reduced heterochromatin formation, which is also a hallmark of aging[8]. Several studies indicate that loss of heterochromatin contributes to genome instability, probably by rendering chromatin more susceptible to DNA damage or by other causes[7,30]. On the other hand, reduction of heterochromatin promotes nuclear softening, which reduces direct force propagation to DNA upon mechanical stress, potentially preventing DNA breaks[31]. Here, we demonstrate an unexpected pivotal role of *Taf4a* for maintaining heterochromatin, genomic stability, and quiescence of MuSC, preventing depletion of the MuSC pool and ensuring skeletal muscle regeneration (Fig. 7).

We found that multiple epigenetic modifiers are differentially expressed in *Taf4a^sKO* compared to WT MuSCs, mostly due to secondary events. The main target of TAF4A in MuSCs turned out to be *Kansl2*, an essential component of the highly conserved NSL complex[32], which is required for acetylation of H4K16, a modification that decompacts chromatin and activates transcription, but also critical for acetylation of lamins. In contrast to most other differentially expressed epigenetic modifiers, the *Kansl2* promoter showed robust binding of TAF4A in CUT&RUN assays and overexpression of *Kansl2* in isolated MuSCs reversed several features of the *Taf4a^sKO* phenotype. Notably, micronuclei formation was prevented in *Taf4a^sKO* MuSCs by directed expression of *Kansl2*, and knockdown of *Kansl2* recapitulated nuclear blebbing and micronuclei formation seen in *Taf4a^sKO* MuSCs (Fig. 7). Numerous promoters of epigenetic modifiers (except *Smyd5* and *Suv39h2*), downregulated in *Taf4a^sKO* MuSCs, showed binding of KANSL2, suggesting that their attenuated expression is caused by reduced activity of the *Kansl2* gene, secondary to the loss of *Taf4a*. The ATAC-seq data revealed chromatin accessibility changes predominantly at intergenic and intronic regions, which are most likely indirect consequences of transcriptional and epigenetic perturbations, rather than direct effects of *Taf4a* inactivation. KANSL2 binds to over 7500 genes, accounting for 46% downregulated transcripts in *Taf4a^sKO* MuSCs, supporting the key role of NSL for the complex phenotype of *Taf4a^sKO* MuSCs. Besides its function in H4K16 acetylation, the NSL complex is well known to mediate acetylation of lamin A/C, which is essential for stability of nuclear architecture[11]. Loss of lamin A/C acetylation impairs nuclear lamina cohesion, causing catastrophic rupture of the nuclear lamina, formation of micronuclei, and genomic instability, a process termed chromothripsis[11,33,34]. We observed all the characteristic features of defective lamin A/C acetylation in *Taf4a^sKO* MuSCs, including the increase of γH2A.X, indicating enhanced DNA damage. Further proof for reduced lamin acetylation came from the strong increase of phospho-Ser 392 lamin A/C and the reduced stiffness of *Taf4a^sKO* MuSCs nuclei. Defective lamin A/C acetylation, partially recapitulates the effects of lamin mutations, which promote loss of heterochromatin at the nuclear periphery (reviewed in ref. 35).

We reason that loss of heterochromatin and increased genomic instability work together to drive MuSC out of quiescence after inactivation of *Taf4a*. DNA damage as a cause of stem cell activation has been described before. Several studies indicate that DNA damage eliminates damaged stem cells by inducing loss of quiescence and promoting differentiation[36]. The degree of DNA damage may be decisive in this context, since stem cells normally cope well with minor defects. For example, murine adult hematopoietic stem cells initiate repair and remain quiescent after low-level DNA damage[37], which changes when DNA repair capacity is compromised. In contrast, deletion of the DNA-damaged activated kinase ATR (Ataxia Telangiectasia and Rad3-Related) in MuSCs enhances MuSC quiescence exit[38]. Inactivation of *Atf3*, required for H2B expression to prevent genome instability, provokes MuSC activation[7]. Apparently, the excessive genome instability in *Taf4a^sKO* MuSCs cannot be healed by repair mechanisms, resulting in MuSC activation and subsequent impairment of proliferation.

TAF4A is best known as a subunit of the basal transcription factor TFIID, composed of the TATA-binding protein (TBP) and 13–14 TBP-associated factors (TAFs), which recruit RNA Polymerase II (RNAPII) to the core promoters of genes[16,39]. However, more recent studies disclosed unexpected roles of TFIID subunits in cell-type-specific transcription during development, differentiation, and cell proliferation[17]. Surprisingly, our CUT&RUN analysis of FACS-sorted MuSCs identified only 1131 TAF4A-binding sites in 928 genes, disclosing a major overlap with SP1 and NF-YA binding sites, but no preferences for the type of core promoter elements. In contrast, TBP shows 13,148 peaks in ChIP-Seq experiments, of which approximately 85% are located in promoter regions[40]. We assume that our CUT&RUN experiments primarily detected TAF4A outside of the TFIID complex, since TAF4A may not easily be detected by antibodies when embedded into the TFIID complex[25,41]. The differential accessibilities for the antibody against TAF4A inside and outside the TFIID complex allowed us to focus on the interaction of TAF4A with NF-Y, which is critical for the physiological function of *Taf4a* in MuSCs.

The collaboration of TAF4A with NF-Y in MuSCs includes the regulation of *Kansl2* expression. Knockdown of *Nf-ya* reduced *Kansl2* expression and disrupted nuclear architecture. These findings correspond well to a recent study, demonstrating that *Nf-ya* is required to preserve MuSCs and enable muscle regeneration[42]. Similar to the loss of *Taf4a*, inactivation of NF-YA-depleted muscle stem cells led to loss of quiescence, increased DNA damage and apoptosis[42]. *Taf4a* clearly exerts cell-type-specific functions, which may depend on the

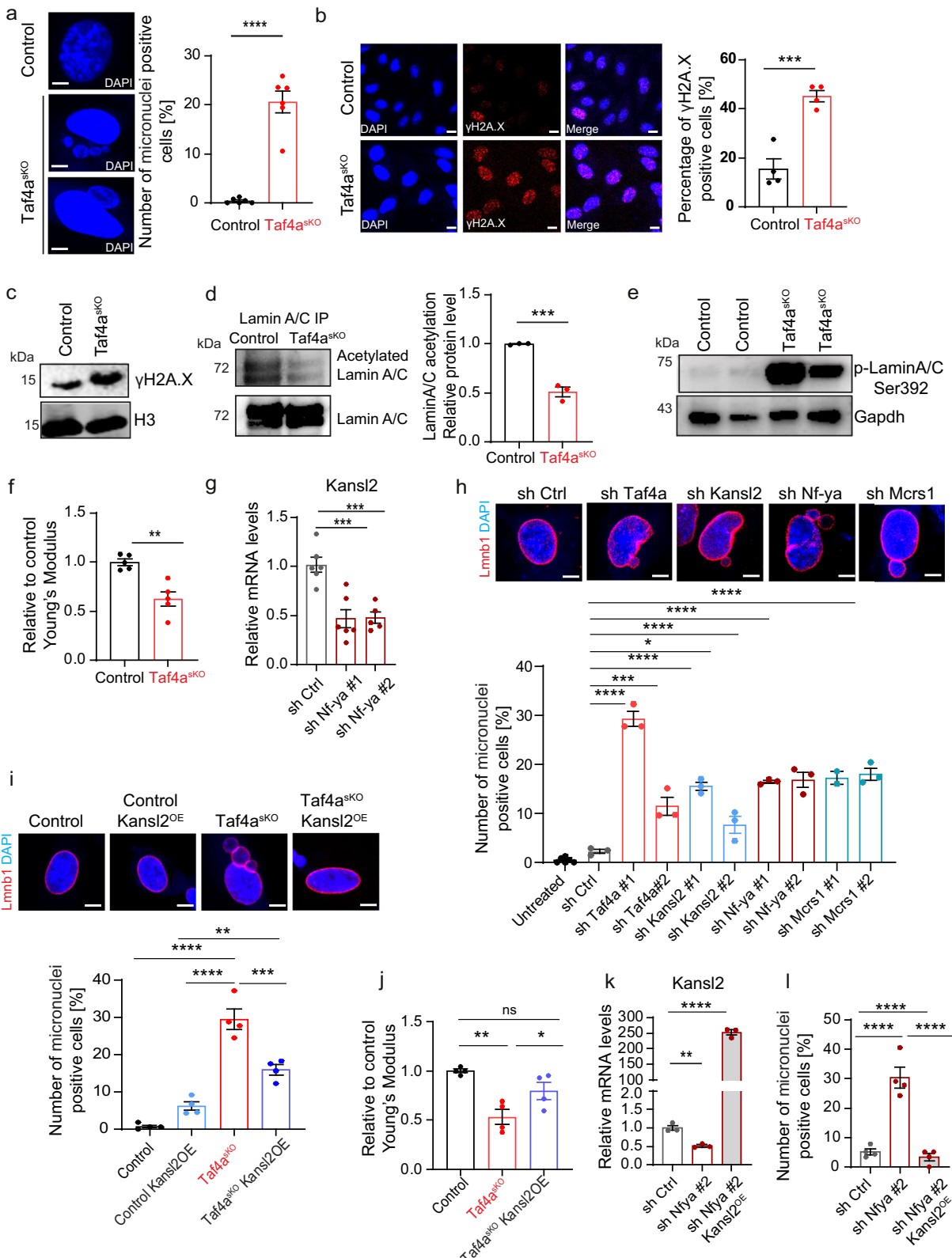

availability of binding partners such as NF-Y. In contrast to MuSCs, we did not detect effects of *Taf4a* depletion on the proliferation of ES cells and mouse embryonic fibroblasts, which is in line with previous reports[19,43]. In the intestinal epithelium, inactivation of *Taf4a* compromises the stem cell compartment, epithelial turnover and differentiation of mature cells, which was reported to be mediated by inhibition of polycomb activity[44]. This mechanism does not seem to be

active in MuSCs, since we did not find increased levels of H3K27me3 in *Taf4a^{sKO}* MuSCs.

The discovery that TAF4A serves a critical role for the proper function of the NSL complex also has important implications for a recently defined group of human pathologies named TAF4-related neuro-developmental disorders (T4NDD). Mutations in *Taf4a* were found in human subjects with neuro-developmental disorders,

**Fig. 5 | Loss of *Taf4a* in adult MuSCs leads to catastrophic genomic instability.**
**a** DAPI staining of $Taf4a^{sKO}$ MuSCs. Quantification of relative micronuclei numbers. Scale bar: 5 μm (****$p < 0.0001$, $n = 6$). **b** gH2A.X immunofluorescence in cultured control and $Taf4a^{sKO}$ MuSCs. Scale bar: 20 μm. Quantification of relative numbers of γH2A.X+ cells (***$p = 0.0008$, $n = 4$). **c** Western blot of γH2A.X in control and $Taf4a^{sKO}$ MuSCs. Loading control: H3 ($n = 3$). **d** Western blot of immunoprecipitated Lamin A/C in control and $Taf4a^{sKO}$ MuSCs. Acetylation signals were normalized to total Lamin A/C levels. Quantification of acetylation levels (***$p = 0.0006$, $n = 3$). **e** Western blot of p-lamin A/C ser392 in cultured control and $Taf4a^{sKO}$ MuSCs. Loading control: GAPDH ($n = 2$). **f** Stiffness measurements of control and $Taf4a^{sKO}$ MuSC nuclei by AFM. Data represent average medians of Young's Modulus (Pa) normalized to control MuSCs (**$p = 0.0017$, $n = 5$). **g** RT-qPCR analysis of *Kansl2* expression in scrambled control and *Nf-ya* knockdown MuSCs. Reference gene: *m36b4* (***$p = 0.0004$, ***$p = 0.0007$, $n = 6$). **h** LaminB staining and quantification of micronuclei-positive cells in untreated ($n = 6$), scrambled control ($n = 3$), *Taf4a* knockdown ($n = 3$), *Kansl2* knockdown ($n = 3$), *Nfya* knockdown ($n = 3$), and *Mcrs1*

knockdown ($n = 3$) in cultured MuSCs (*$p = 0.0214$, ***$p = 0.0001$, ****$p < 0.0001$). Scale bar: 5 μm. **i** LaminB staining and quantification of micronuclei-positive cells in control, *Kansl2* overexpressing control, $Taf4a^{sKO}$ and *Kansl2* overexpressing $Taf4a^{sKO}$ MuSCs (**$p = 0.0054$ ***$p = 0.0004$ ****$p < 0.0001$, $n = 4$). Scale bar: 5 μm. **j** Stiffness measurements of control, $Taf4a^{sKO}$ and *Kansl2* overexpressing $Taf4a^{sKO}$ MuSC nuclei by AFM. Data representation as in (**f**) (**$p = 0.0026$, *$p = 0.0437$ $n = 4$). **k** RT-qPCR analysis of *Kansl2* expression in scrambled control, *Nf-ya* knockdown and *Nf-ya* knockdown in *Kansl2* overexpressing MuSCs. Reference gene: *m36b4* (**$p = 0.0026$, ****$p < 0.0001$, $n = 3$). **l** Quantification of micronuclei-containing cells in scrambled control, *Nf-ya* knockdown, and *Nf-ya* knockdown *Kansl2* over-expressing MuSCs (****$p < 0.0001$, $n = 4$) from 8-20 weeks old males and females, isolated 4–6 weeks after tamoxifen administration. Data are mean ± SEM of biological replicates. (**a**, **b**, **d**, **f**, **k**) unpaired two-tailed $t$ test. (**g**, **h**, **l**) one-way ANOVA with Bonferroni's multiple comparisons test. (**i**) one-way ANOVA with Tukey's multiple comparisons test. (**j**) one-way ANOVA with Holm−Sidak's multiple comparisons test. Source data are provided in the Source Data file.

including features such as intellectual disability, abnormal behavior, and facial dysmorphisms[45]. We expect that affected individuals will also carry skeletal muscle defects, which so far might have been sidelined by the neurological symptoms. Our finding that TAF4A interacts with the NF-Y transcription factor in MuSCs to regulate expression of *Kansl2* for maintaining stability and function of the NSL complex offers new opportunities to study T4NDDs and develop novel therapeutic approaches.

## Methods

### Animals
*Taf4a* mice were generated by flanking exon 11 to 12 of the *Taf4a* gene with two loxP sequences. The mouse line was kindly provided by Prof. Irwin Davidson. *Pax7Cre* and *Pax7-zsGreen* mice have been described previously[46]. Primers used for genotyping are listed in (Supplementary Table 1). Tamoxifen (Sigma) was administered to 8–20-week-old mice intraperitoneally at 0.05 mg/g of body weight per injection. All animal experiments were done in accordance with the Guide for the Care and Use of Laboratory Animals published by the US National Institutes of Health (NIH Publication No. 85-23, revised 1996) and according to the regulations issued by the Committee for Animal Rights Protection of the State of Hessen, Regierungspraesidium Darmstadt, with the project numbers B2/1137 and B2/2048.

### Muscle regeneration assay
Mice were anaesthetized intraperitoneally with 10% ketamine (Bela-pharm) and 2% xylazine (WDT eG, Garbsen, Germany) diluted with 0.9% NaCl (100 μl /10 g body weight). To introduce a muscle injury, 50 μl of 0.06 mg/ml cardiotoxin from Naja mossambica mossambica (Sigma, Germany) in 0.9% NaCl was injected into the tibialis anterior of adult mice using an insulin syringe. The needle was inserted deep into the muscle longitudinally towards the knee from the ankle. The anterior tibial muscles were analyzed 14 days after injection.

### Isolation and cultivation of MuSC
Satellite cell isolation and purification were performed according to established methods[47] with minor modifications. Briefly, limb and trunk muscles were minced, digested with 100 CU Dispase (BD) and 0.2% type II collagenase (Worthington Biochemicals), and consecutively filtered through 100 μm, 70 μm, and 40 μm cell strainers (BD). Cells were collected by centrifugation at 1200$g$ for 7 min. Pellets were re-suspended in 1.5 ml red blood cell lysis buffer containing 5 μg/mL DNase I and incubated on ice for 3 min. Subsequently, the cell suspension was filled up to 7 ml with DMEM medium containing 2% FCS, before cells were spun down. To enrich for MuSCs, isolated cells were incubated with APC fluorescence coupled primary antibodies against SCA1, CD45, CD31 (1:100 dilution in FACS sorting buffer) for

40 min at 4 °C. After addition of 5 ml of DMEM medium containing 2% FCS, cells were spun down and the cell pellets were resuspended in 200 μl FACS sorting buffer, before incubation with 30 μl of anti-APC micro beads (MACS) for 15 min on 4 °C. Microbeads containing SCA1 + / $CD45^+/CD31^+$ cells were isolated by a 25 LS separation column using the QuadroMACS separator (Miltenyi Biotec). $SCA1^-/CD45^-/CD31^-$ cells were spun down, stained with Integrin-a7-FITC antibody (1:100 dilution in FACS sorting buffer) for 1 hour. Integrin-a7$^+$ or GFP$^+$ (from *Pax7-zsgreen* mice) satellite cells were isolated using a FACS AriaIII (BD Biosciences) (Supplementary Fig. 7). Antibodies used for FACS-sorting are listed in (Supplementary Table 2). FACS-sorted MuSCs were cultured on Matrigel-coated or 96-wells or 384-wells μClear plates (BD Biosciences, Greiner) in containing high-glucose DMEM GlutaMAX™ (Gibco) with 20% FCS, LIF (10 ng/ml) and bFGF (5 ng/ml).

### Myofiber Isolation and Culture
Flexor digitorum brevis (FDB) muscles were dissected from adult mice and digested with 0.2% collagenase P (Roche) in high-glucose DMEM for 1 hour at 37 °C. Following digestion, individual myofibers were released by gentle trituration. Fibers were then transferred to 6-cm culture dishes containing high-glucose DMEM GlutaMAX™ (Gibco) supplemented with 10% horse serum (Gibco) and 1% chick embryo extract (MP Biomedicals Europe), and maintained under floating conditions at 37 °C and 5% $CO_2$ for 3 days.

### Gene expression and RT-qPCR
Total RNA from FACS-isolated MuSCs was extracted using TRIzol reagent (Invitrogen) following instructions of the manufacturer. 1 μg purified RNA was reverse-transcribed with PrimeScript RT Reagent Kit (TaKaRa) following standard procedures. Real-time PCR was performed using Blue S´Green qPCR Kit (Biozym Blue). Relative quantitation of mRNA gene expression was performed using the ΔCT method. The Ct-values of target genes were normalized to the *m36b4* housekeeping gene using the equation $\Delta Ct = Ct_{reference} - Ct_{target}$ and expressed as ΔCt. Primers used for RT-qPCR are listed in (Supplementary Table 1).

### Co-immunoprecipitation
C2C12 cells (ATCC #CRL-1772) or primary MuSCs were washed twice with ice-cold PBS, scraped, and pelleted by centrifugation. Cell pellets were lysed in RIPA buffer (50 mM Tris-HCl, pH7.4; 150 mM NaCl; 1% NP-40; 0.1% sodium deoxycholate; complete protease inhibitor cocktail (Roche)) and briefly sonicated. Lysates were pre-cleared by incubation with Protein A/G agarose beads (Sigma Aldrich) for 1 hour at 4 °C with rotation. Pre-cleared lysates were centrifuged at 15,000 g for 10 minutes at 4 °C. The supernatant was collected and incubated with 6 μg anti-TAF4A antibody (Santa Cruz) overnight at 4 °C on a rotating

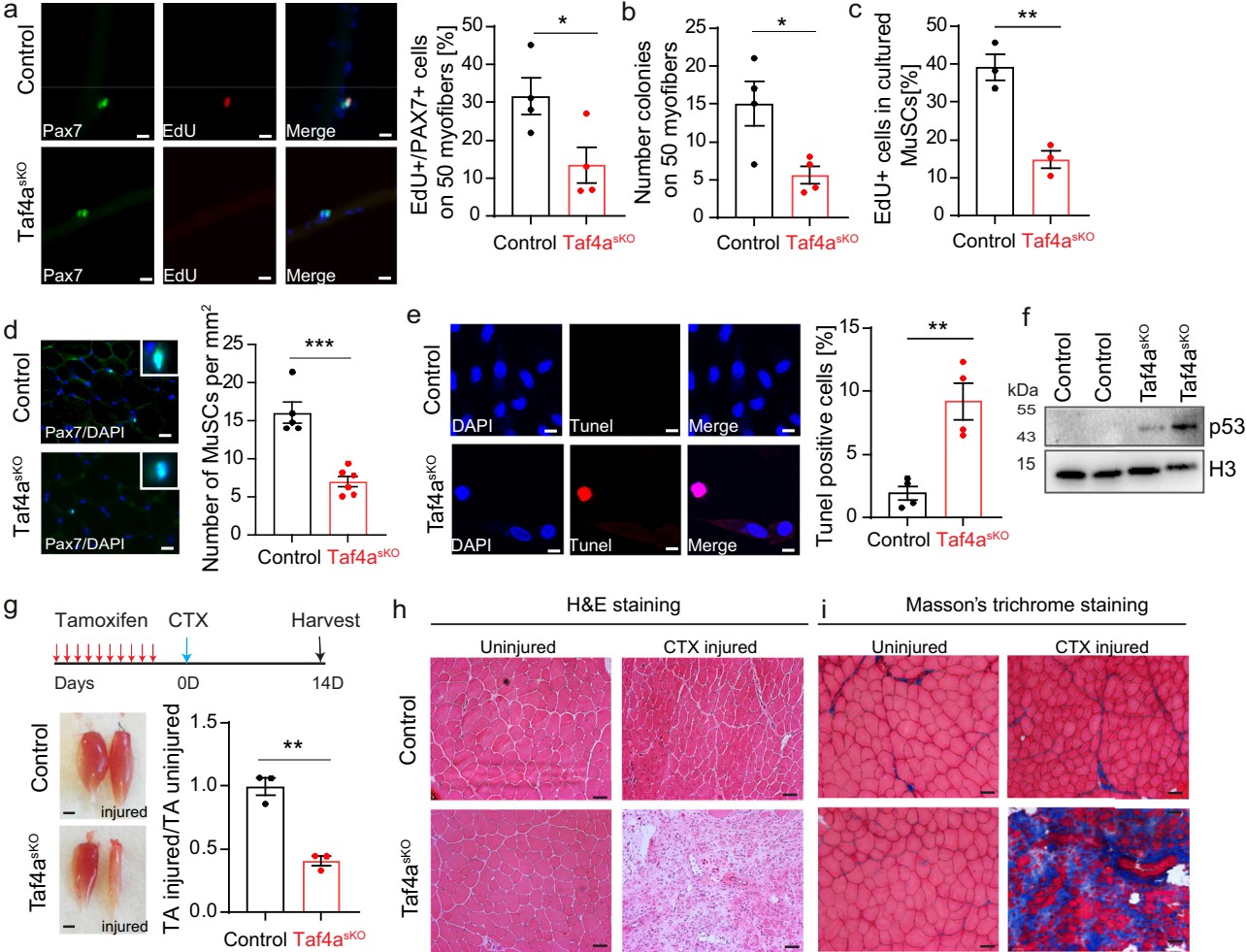

**Fig. 6 | Inactivation of *Taf4a* abrogates skeletal muscle regeneration.**
**a** Immunofluorescence staining of EdU/PAX7 double-positive MuSCs on FDB myofibers from control and *Taf4a^sKO^* mice, cultured for 3 days. Scale bar: 20 μm. Quantification of relative numbers of EdU + /PAX7+ cells found on 50 myofibers (unpaired two-tailed *t* test: * *p* = 0.0377, *n* = 4). **b** Quantification of Pax7+ colonies per 50 myofibers (unpaired two-tailed *t* test: **p* = 0.0244, *n* = 4). **c** Quantification of EdU-incorporation in cultured control and *Taf4a^sKO^* MuSCs (unpaired two-tailed *t* test: ***p* = 0.0044, *n* = 3). **d** Immunofluorescence staining of PAX7-positive MuSCs in control (*n* = 5) and *Taf4a^sKO^* (*n* = 6) TA muscle sections. Scale bar: 20 μm. Quantification of PAX7+ cells per mm² section area is in the right panel (unpaired two-tailed *t* test: ****p* = 0.0002). **e** TUNEL assay of cultured control and *Taf4a^sKO^* MuSCs. Scale bar: 10 μm. Quantification of TUNEL-positive cells is shown on the right

(unpaired two-tailed *t* test: ***p* = 0.0034, *n* = 4). **f** Western blot analysis of p53 expression in cultured MuSCs from control and *Taf4a^sKO^* mice. H3 was used as loading control (*n* = 2). **g** Outline of the muscle regeneration assay. Representative macroscopic views of TA muscles from control and *Taf4a^sKO^* mice 14 days after muscle injury. Scale bar: 1 mm. Ratio of TA muscle masses with and without injury in control and *Taf4a^sKO^* mice (unpaired two-tailed *t* test: ***p* = 0.0017, *n* = 3). **h, i** H&E (**h**) and Trichrome staining (**i**) of TA muscle sections from control and *Taf4a^sKO^* mice 14 days after injury (*n* = 3). Scale bar: 20 μm. Myofibers and MuSCs were isolated 4–6 weeks after tamoxifen administration in (**a**–**f**). Muscles were isolated 2 weeks after tamoxifen administration in (**g**–**i**). 8–20-week-old males and females were used. Data are presented as mean ± SEM of biological replicates. Source data are provided in the Source Data file.

wheel. The next day, 30 μL Dynabeads Protein G (Invitrogen) were added and incubated for 2 hours at 4 °C. Beads were washed three times with RIPA buffer, resuspended in SDS loading buffer, and analyzed by Western blotting.

## Lamin A/C Immunoprecipitation

Cells were washed twice with ice-cold PBS, scraped and pelleted by centrifugation. The cell pellet was resuspended in hypotonic buffer (10 mM HEPES pH 7.9, 10 mM KCl, 1.5 mM MgCl₂, 0.5 mM DTT, and protease inhibitors) and incubated on ice for 5 minutes. Nuclei were isolated by centrifugation at 1,500 × g for 5 minutes at 4 °C. The nuclear pellet was then resuspended in lysis buffer containing 20 mM Tris-HCl (pH7.4), 200 mM NaCl, 2% Triton X-100, benzonase (25–50 U/mL, Sigma-Aldrich), complete protease inhibitor cocktail (Roche) and histone deacetylase inhibitors (1 μM TSA and 5 mM NAM). Samples were briefly sonicated (10 pulses, low intensity) and incubated on a

rotating wheel at 4 °C for 2 hours. Lysates were clarified by centrifugation at 15,000 g for 10 minutes at 4 °C and the supernatant was collected. Three micrograms of anti-Lamin A/C antibody (Santa Cruz) was added and incubated overnight at 4 °C a rotating wheel. The next day, 20 μL of Dynabeads Protein G (Invitrogen), pre-washed and equilibrated in lysis buffer, was added for 2 hours at 4 °C. Beads were washed three times with the same lysis buffer (200 mM NaCl, 2% Triton X-100), using a magnetic stand to collect the beads between washes. The flow-through from the first IP was retained, and the immunoprecipitation procedure was repeated once more to improve yield. Beads were finally resuspended in SDS sample buffer for further immunoblot analysis.

## Western blot analysis

Cell pellets from freshly isolated or cultured MuSCs were lysed in cell lysis buffer (20 mM Tris pH 7.5, 400 mM NaCl, 1 mM EDTA, 1 mM EGTA,

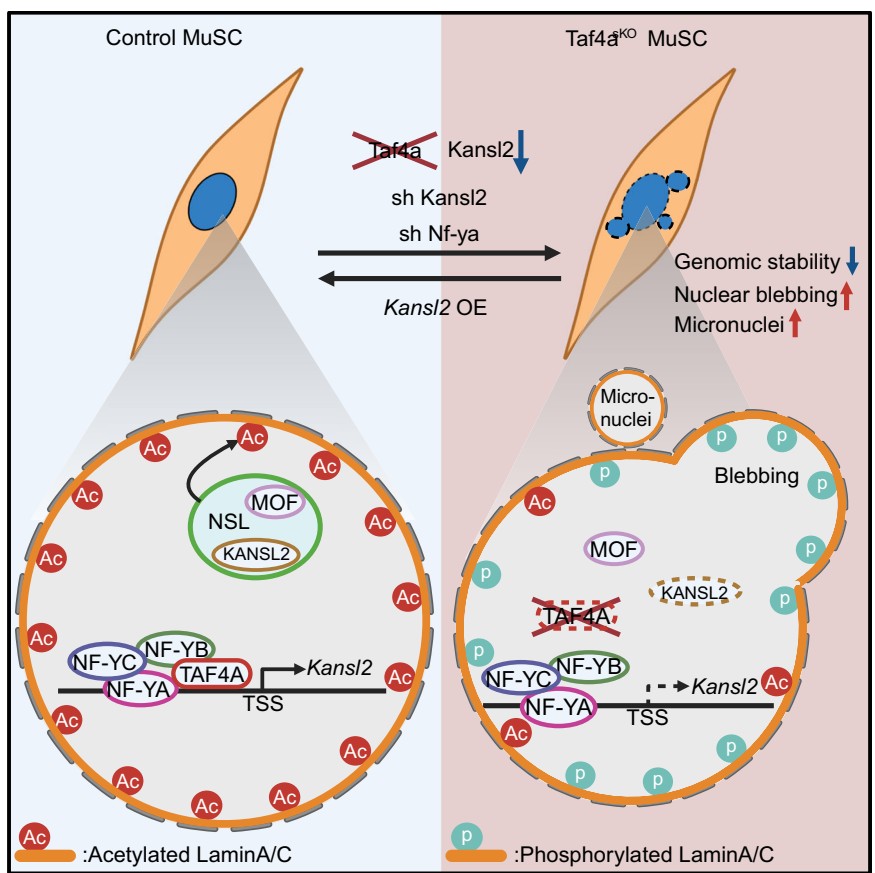

**Fig. 7 | TAF4A protects genome stability of MuSCs by enabling NSL-dependent acetylation of lamin A/C.** TAF4A drives expression of *Kansl2* by interacting with the NF-Y complex. KANSL2 is a critical part of the NSL (non-specific lethal) complex, which also comprises the acetyl-transferase MOF. Decreased expression of *Kansl2* prevents NSL-mediated acetylation of Lamin, increasing Lamin phosphorylation and solubility. Created in BioRender. Guo, X. (2025) https://BioRender.com/frmva1q.

1% Triton X-100, 2.5 mM Sodium pyrophosphate, 1 mM β-glycerophosphate, 1 mM Na3VO4, 1 µg/ml Leupeptin). Extracts from MuSCs were resolved by SDS-PAGE, transferred onto nitrocellulose filters or PVDF membranes and probed with antibodies as indicated (Supplementary Table 2). Protein expression was visualized using a chemiluminescence detection system (GE Healthcare, Little Chalfont, United Kingdom) or imaged using the LI-COR Odyssey system (BioAgilytix) and quantified with ImageJ.

### Immunofluorescence and histological analysis
Tibialis anterior muscles were dissected and placed with one end in 10% gum tragacanth (Sigma-Aldrich) on a flat piece of cork before freezing in isopentane/liquid nitrogen. Frozen muscle sections (8-10 µm) were fixed in cold acetone or 4% paraformaldehyde. Immunofluorescence and H&E staining were done following standard protocols. Masson's Trichrome staining was carried out using the ACCUSTAIN® trichrome staining kit (Sigma-Aldrich). Primary antibodies for immunofluorescence staining are listed in (Supplementary Table 2). MuSCs were identified by staining for PAX7. For quantification, the numbers of PAX7-positive MuSC were counted on 5 sections per animal, using 4 different areas per section.

### EdU labeling
For in vitro labeling experiments, MuSCs or myofibers were treated with 10 µM 5-ethynyl-2′-deoxyuridine for the last 3 hrs of culture. Cells were fixed 4%PFA, permeabilized, and EdU incorporation was detected using the Click-iT EdU Alexa Fluor 594 Imaging Kit (Life Technologies) according to the manufacturer's protocol.

### TUNEL assay
Cells were fixed in 4% PFA for 10 min and then subjected to the TUNEL reaction using the TUNEL Assay Apoptosis Detection Kit (C10246, Invitrogen), following the manufacturer's instructions.

### Lentiviral transduction of MuSCs, mESCs and NIH3T3
FACS-sorted MuSCs were cultured in DMEM medium with 20% FCS and bFGF (5 ng/ml). Before seeding of MuSCs, plates or dishes were coated with matrigel at 37 °C for 1 hour and then air-dried. 1000 cells/well were seeded into a 384-well plate and 2×10⁴ cells/well in 12-well plate for IF assay and RT-qPCR, respectively. On the second day after seeding, cells were transduced with lentiviruses containing shRNA against *Taf4a*, *Kansl2*, *Mcrs1*, or *Nf-ya* and/or lentiviruses over-expressing *Kansl2* (Supplementary Table 3 and 4). Lentivirus mediated *Taf4a* knockdown in mESC (V6.5 NBP1-41162) and NIH3T3 (ATCC CRL-1658) was performed with shRNA shown in Supplementary Table 3, following standard protocols (The RNAi Consortium (TRC) Broad Institute). Briefly, mESCs were harvested and the cell pellet was re-suspended in lentivirus-containing medium with 8 µg/ml polybrene, before plating on a monolayer of feeder cells. After 24 hours incubation, the medium was replaced with growth media containing 2 µg/ml puromycin for another 4 days before further analysis.

### Electron microscopy (EM) and heterochromatin quantification
Ultrastructure of MuSC in skeletal muscle was analyzed by EM as described before[47]. Briefly, skeletal muscles were fixed in 3% glutaraldehyde for 12 hours at 4 °C and embedded in Epon. Ultrathin sections were contrasted with uranyl acetate and lead citrate, and

analyzed using a Philips CM10 electron microscope. The content of heterochromatin was determined using the ImageJ program relative to the total cross-sectional area of MuSC nuclei.

## Atomic force microscopy (AFM)

AFM-based force indentation spectroscopy was performed on proliferative MuSCs cells, after FAC-sorting and culture for 4 days. AFM measurements were performed using the JPK NanoWizard 4XP (Bruker Nano) atomic force microscope mounted on a Zeiss Axio Observer optical inverted fluorescent microscope and operated via JPK SPM Control Software v.7. Triangular non-conductive Silicon Nitride cantilevers (MLCT, Bruker Daltonics) with a nominal spring constant of 0.07 Nm−1 were used for the nanoindentation experiments of the nucleus. For all indentation experiments, forces of up to 1 nN were applied, and the velocities of cantilever approach and retraction were kept constant at 10 μm s$^{-1}$, ensuring an indentation depth of 500 nm. All analyses were performed with JPK Data Processing Software (Bruker Nano; v 7.0.165). Prior to fitting the Hertz model corrected by the tip geometry to obtain Young's Modulus (Poisson's ratio of 0.5), the offset was removed from the baseline, the contact point was identified, and cantilever bending was subtracted from all force curves.

## RNA-Seq and data analysis

Total RNA was isolated from FACS-sorted MuSCs by the miRNeasy micro kit (Qiagen) together with on-column DNase digestion (DNase-Free DNase Set, Qiagen) to avoid contamination by genomic DNA. RNA and library preparation integrity were verified with LabChip Gx Touch (Perkin Elmer). 10 ng of total RNA was used as input for SMARTer® Stranded Total RNA-Seq Kit v2 - Pico Input Mammalian. Library preparations followed the manufacturer's protocol (Takara Bio). Sequencing was performed on a NextSeq500 instrument (Illumina) using a P3 flowcell with 75 bp single-end setup, resulting in an average of 30 M reads. Trimmomatic version 0.39 was employed to trim reads after a quality drop below a mean of Q15 in a window of 5 nucleotides and keeping only filtered reads longer than 15 nucleotides[48]. Reads were aligned to the Ensembl mouse genome version mm10 (Ensembl release 101) with STAR 2.6.0c[49]. Alignments were filtered to remove multi-mapping reads. Gene counts were established with feature-Counts 1.6.0 by aggregating reads overlapping exons, excluding those overlapping multiple genes[50].

The raw count matrix was normalized with DESeq2 version 1.14.1[51]. Only genes with a minimum log fold change of ±0.378, a maximum Benjamini-Hochberg corrected p-value of 0.05, and a minimum combined mean of 5 reads were assumed to be significantly differentially expressed. Identified DEGs were uploaded to the online software DAVID and EnrichR for GO or KEGG pathway analyses.

## CUT&RUN (CnR) and data analysis

CnR was performed as described previously[52]. In brief, $1 \times 10^6$ wild type in vitro cultured muscle stem cells (MuSCs) or $4 \times 10^5$ wild type and *Taf4a*$^{sKO}$ MuSCs were immobilized on Concanavalin A-coated magnetic beads (Bangs Laboratories), permeated with 0.05% Digitonin (EMD Millipore), and incubated with TAF4A (Santa Cruz #sc-136093), KANSL2 (Proteintech #27261-1-AP), NF-YA (Santa Cruz #sc-17753) or mouse IgG (Millipore #12-371B) or rabbit IgG (Diagenode #C15410206) on a rotator with 1:100 dilution at 4 °C overnight on a rotator. After washing, the beads were incubated with home-made pA-MNase (0.3 ng/μl final concentration) at room temperature for 1 h and DNA was cleaved by incubation with buffer containing 10 mM CaCl$_2$ and 3.5 mM HEPES pH 7.5 at 0 °C for 20 min. Under this high-calcium/low-salt condition, cleaved DNA fragments were released and then extracted using phenol-chloroform.

DNA was quantified by Qubit and max. 10 ng DNA was used as input for SMARTer® ThruPLEX® DNA-seq Kit following the manufacturer's protocol (Takara Bio). Sequencing was performed either on the NextSeq500 platform (Illumina) using a P3 flowcell with a 2x38bp single-end setup or on the NextSeq2000 platform (Illumina) using a P3 flowcell with a 2x61bp single-end setup or 2x36bp. The following steps are the same as described previously[13]. Peak calling was performed with Macs version 2.1.1 with FDR < 0.05 and additional parameters "--extsize 100 --nomodel --to-large"[53] (GSE277867, GSE277868, GSE277869) and Macs version 3.0.0a7 with FDR < 0.1 and additional parameters "--extsize 75 --nomodel --to-large" (GSE277870). The remaining peaks were unified to represent a common set of regions for all samples. Counts were produced with featureCounts[50]. The raw count matrix was normalized with DESeq2 version 1.30.0[51]. Peaks were annotated with the promoter (TSS + − 10000 nt) of the nearest gene based on Ensembl data. Genomic tracks of sequencing data were visualized using IGV.

## ATAC-Seq

For ATAC library preparation, 50.000 cells were employed, using the Tn5 Transposase from Nextera DNA Sample Preparation Kit (Illumina). Cell pellets were resuspended in 50 μl PBS and mixed with 25 μl TD-Buffer, 2.5 μl Tn5, 0.5 μl 10% NP-40 and 22 μl water. Cell/Tn5 mixture was incubated at 37 °C for 30 min with occasional snap mixing. Transposase treatment was followed by 30 min incubation at 50 °C together with 500 mM EDTA pH8.0 for optimal recovery of digested DNA fragments. 100 μl of 50 mM MgCl2 was added for neutralization of EDTA, followed by purification of DNA fragments by the MinElute PCR Purification Kit (Qiagen). Amplification of the library together with indexing was performed as described elsewhere[54]. Libraries were mixed in equimolar ratios and sequenced by the NextSeq500 platform using V2 chemistry with paired-end mode. Trimmomatic version 0.39 was employed to trim reads after a quality drop below a mean of Q15 in a window of 5 nucleotides and keeping only filtered reads longer than 15 nucleotides[48]. Reads were aligned versus Ensembl mouse genome version mm10 (Ensembl release 101) with STAR 2.7.11b[49]. Alignments were filtered to remove duplicates with Picard 3.1.1 (http://broadinstitute.github.io/picard), including spliced, multi-mapping, ribosomal, or mitochondrial reads. Peak calling was performed with Macs version 3.0.0a7 with FDR < 0.0001[53]. The remaining peaks were unified to represent a common set of regions for all samples. Counts were produced with featureCounts[50]. The raw count matrix was normalized for sequencing depth. Peaks were annotated with the promoter (TSS + − 10000 nt) of the nearest gene based on Ensembl release 101. Contrasts were created with DESeq2 based on the normalized count peak matrix with all size factors set to one. Peaks were classified as significantly differential at average count > 10 and −1 <log2FC > 1.

## Statistical analysis

For all quantitative analyses, a minimum of three biological replicates were analyzed. Statistical tests were used based on the assumption that sample data are derived from a population following a probability distribution based on a fixed set of parameters. T-tests were used to determine the statistical significance of differences between two groups. One-way ANOVA was used to perform multiple comparisons. The following values were considered to be statistically significant: $*P < 0.05$, $**P < 0.01$, $***P < 0.001$, $****P < 0.0001$ and exact P-values are written in the figure legend. Calculations were done using the Graph-Pad Prism 9 software and R. Error bars always represent the standard error of the mean (mean ± SEM). No statistical method was used to predetermine sample size.

## Reporting summary

Further information on research design is available in the Nature Portfolio Reporting Summary linked to this article.

## Data availability

Raw and processed CUT&RUN data for TAF4A, NF-YA and KANSL2 in wild type MuSCs, ATAC-seq data, NF-YA CUT&RUN data in control and *Taf4a*^sKO MuSCs, and RNA-seq data of freshly isolated WT and *Taf4a*^sKO MuSCs are available in the NCBI Gene Expression Omnibus (GEO), under accession number GSE277872. Published H3K4me3 ChIP-seq data were downloaded from GEO time-point T3 from "GSE103163" GSM2756400 and GSM2756401. Published RNA-seq data of freshly isolated and activated/proliferating WT MuSCs were downloaded from GEO under accession number "GSE108040". Published RNA-seq data of freshly isolated WT MuSCs and WT muscle were downloaded from GEO under accession number "GSE199487" and GSE168984, respectively. Published RNA-seq data of WT and NF-YA knockout MuSCs were downloaded from GEO under accession number "GSE154017". Source data are provided with this paper.

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

## Acknowledgements

The authors are sincerely grateful to Prof. Irwin Davidson (Institute for Genetics, Molecular and Cellular Biology, Strasbourg) for providing Taf4a floxed mice. We thank Kikhi Khrievono, Ann Atzberger, and Martin Stehling for support with FACS sorting. We would also like to thank Birgit Spitznagel and Barbara Zimmermann for technical help. This work was supported by the Excellence Cluster Cardio Pulmonary System (CPI), the Deutsche Forschungsgemeinschaft (DFG) Collaborative Research Centre 1213 (TP AO2 and BO2) and 1531 (TP BO8), the DFG Transregional Collaborative Research Center 267 (TP A05), the LOEWE project iCANx.

## Author contributions

A.M.G., X.Y., and T.B. conceived and designed this project. AMG performed most of the experiments, analyzed the data, and prepared figures. K.S. performed knockdown experiments. D.D. helped with CUT&RUN experiments. C.V. and S.A.W. performed atomic force microscopy (AFM) and analyzed the data. S.G. and C.K. analyzed RNA-seq and CUT&RUN data. U.G. performed electron microscopy imaging. X.G. performed co-IP experiments. Y.Z. contributed to data analysis, discussion and advice. A.M.G., X.Y., and T.B. wrote the manuscript.

## Funding

## Competing interests

The authors declare no competing interests.
