## [Transparent Peer Review file · Nature Communications]

Regulation of NSL by TAF4A is critical for genome stability and quiescence of muscle stem cells

Corresponding Author: Professor Thomas Braun

Version 0:

Reviewer comments:

Reviewer #1

(Remarks to the Author)

Georgieva et al. report that TAF4A is critical for genome stability and quiescence of skeletal muscle stem cells.

The findings are of interest and the experiments well-executed and controlled.

However, the following points should be addressed before publication.

1. To avoid unintended MuSC activation, staining of uninjured control and TAF4asKO muscle sections should be performed to evaluate MyoD protein accumulation.
2. ATAC-seq would determine whether heterochromatin reduction occurs at specific genomic regions and whether this phenomenon is associated with modified gene expression.
3. Decreased H3K9me3 is expected to favor gene activation. Was H3K9me3 reduced at the epigenetic modifiers downregulated in TAF4asKO MuSCs? Is reduced H3K9me3 the result of indirect effects of TAF4A deletion?
4. Figure 3f: In addition to overexpressed, endogenous TAF4A-NFYB interaction should be documented.
5. Were the NF-YA occupied genes active in TAF4asKO MuSCs? If they were, what is the functional significance of NF-YA-TAF4A interaction at the approximately 700 genes (Figure 3g) bound by both NF-YA and TAF4A?
6. H3K9me3 heterochromatin mediates nuclear stiffness (Nava et al. Cell 2020, <https://doi.org/10.1016/j.cell.2020.03.052>). To evaluate whether stiffness reduction of nuclei in TAF4asKO MuSCs is due to a direct consequence of reduced KANSL2 rather than indirect effects, TAF4asKO MuSCs may be reconstituted by expressing exogenous KANSL2 followed by AFM measurements.

Reviewer #2

(Remarks to the Author)

The study by Georgieva et al., titled "Regulation of the NSL complex by TAF4 is critical for genome stability and quiescence of skeletal muscle stem cells" investigated the role of TATA-binding protein-associated factor, TAF4A, in the maintenance of heterochromatin and quiescent state of muscle stem cells (MuSCs). The authors demonstrated that TAF4A together with NF-YA regulate the expression of KANSL2. They further showed that KANSL2 as a component of the NSL complex, regulates the expression of key epigenetic modifiers in MuSCs. Using CUT&Run the authors also showed the enrichment of TAF4A and KANSL2 at various genes including those involved in cell cycle control and DNA replication, p53 pathway and apoptosis. Loss of TAF4A results in perturbed TAF4A-NFY-NSL axis, leading to genomic instability with increased gH2A.X protein and foci, and nuclear blebbing. The authors further demonstrated that loss of Taf4a in MuSCs causes atypical activation of MuSCs with impaired proliferation and increased apoptosis, ultimately leading to compromised muscle regeneration.

Overall, the study has demonstrated the significance of heterochromatin and genomic stability via TAF4A-NFY-NSL epigenetic axis during MuSC maintenance. However, at places the authors did not provide strong evidence, and the interpretations are less substantiated. The manuscript could benefit from revisions on the following points.

Major Comments:

1. The manuscript has strongly emphasized the structural integrity of Lamin A/C, specifically by post-translational modifications. However, the authors did not present any evidence on change in the acetylation status of lamins in the absence of TAF4A, except the phosphorylation. The authors must show the differential solubility of lamins (refer to PMID: 31576060) in Taf4asKO and Kansl2 knockdown or can immunoprecipitate with acetyl-lysine and probe with lamin.
2. In figure 2, comparative RNA-seq data, it is important to mention at what time point the activated and proliferating WT MuSCs were collected for comparing expression profiles with freshly isolated Taf4asKO. The authors also need to provide an analysis of the 428 genes that are differentially downregulated in Taf4asKO but upregulated in aMuSCs to understand and explain the phenotype of Taf4asKO MuSCs. Similarly, the authors need to provide a separate analysis, including GO dot plot (to understand the percent genes associated with a term) where they compared freshly isolated WT MuSCs with freshly isolated Taf4asKO MuSCs.
3. The authors showed the interaction of NF-YB with TAF4A and then went on to show the CUT&Run experiments with NF-YA. The authors should discuss about NF-YA/TAF4A interaction, Also, the interaction is shown in HEK293T, which is less significant due to the absence of any nuclear phenotype in embryonic fibroblasts. The authors must show the interaction of NF-YA/NF-YB with TAF4A, in either MuSCs or C2C12 myoblasts.
4. Heatmaps and genomic distribution of NF-YA need to be provided. Among the 48 NF-YA peaks lost in Taf4asKO MuSCs, how many of these are differentially expressed in Taf4asKO MuSCs.
5. In figure 2, the authors reported that among the 21 genes including Smyd5 and Suv39H2, are potential direct targets of TAF4A. However, in Figure 4i, the authors mention about Kansl2 as a regulator of these epigenetic modifiers. Surprisingly Smyd5 and Suv39H2 are missing in this gene list. It is important to dissect which of these are TAF4A and Kansl2 targets. The authors can show that peak distribution for TAF4A and Kansl2 at these genes.

Minor Comments:

1. In the figure 1e, the MyoD1 staining is less convincing due to non-specific background. The authors should use other antibodies to present this data.
2. In figure 1B and 1C, the authors have a clearly shown a decrease in H3K9me3. Redundantly, the authors showed the same decrease in K9 trimethylation in the figure 2e and 2f. Though the authors were trying to corroborate that the decrease in methylation is due to Smyd5 and Suv39h2, it is less informative. Immunofluorescence panels are missing scales.
3. It will be informative to show all the GO plots in a dot plot to understand the number of genes associated with a term.
4. In figure 6a is not truly representing the plotted graph. 3-days post isolation, there are very few MuSCs on the fiber. Similarly, figure 6d is not truly representing the graph. The authors should consider providing representative images.

Reviewer #3

(Remarks to the Author)

A very interesting study demonstrating that TAF4A-NF-Y safeguards heterochromatin and genome stability of skeletal muscle satellite cells via the NSL complex. This has relevance not only for basic muscle biology but also for TAF4-related neuro-developmental disorders in human, which have not been reported to have direct muscular involvement. I recommend publication in Nature Communications.

Major Comment

- Other evidence is supportive (lamin phosphorylation changes, nuclear stiffness and structural changes) but is it possible to also directly show altered acetylation of lamin as a consequence of Taf4a knockout to further support the mechanism?

Other Comments

- Might be useful to state the non-specific lethal (NSL) complex on first mention in the abstract.
 - Would include 'satellite cell' when defining skeletal muscle stem cells (MuSCs) for clarity as several other cell types have been proposed as muscle stem cells over the years. Also, 'satellite cell' is only mentioned in the material and methods otherwise.
 - Graphs in general have enough room to place X axis labels of 'Control' and 'TAF4a sKO' under appropriate column – TAF4a sKO text in red rather than dots?
 - Could comment on how well the NSL complex is conserved in human.
 - No description in material and methods of how muscle fibres were isolated.
 - If possible, including a model would be useful (graphical abstract format).
-
- Figure 1: Here and elsewhere, could n=3 be explained: assume it is n=3 mice but how many satellite cells were examined in Fig. 1a per mouse?
 - Supplementary Figure 1c: TAF4a immunolabeling in wt PAX7-containing satellite cells should instead be in Figure 1 as data presented is fundamental to the paper. Would suggest a higher power image of a satellite cell on a muscle fibre with surrounding myonuclei to better show the higher TAF4a protein levels in the satellite cell.
 - Figure 1c: bands look a bit strange e.g. H3K9me3 in Taf4a sKO. Here and elsewhere (e.g. Fig 2f, 4e, 5c,d, 6f), where just bands excised from a gel are presented, the whole gel should be shown in each case as Supplementary data.

- Figure 1f: calcitonin receptor (CalcR) expression not only labels quiescent satellite cells (e.g. doi:10.1371/journal.pone.0005205).
- Figure 3h: would indicate in figure legend that CnR is CUT&Run.
- Figure 4b: why are changes in others factors so inconsistent?
- Figure 6: using the isolated myofibre assay, it would be informative to know the fate of the associated satellite cells in the Taf4a sKO after 3 days. Is there change in the ratio of those that differentiate versus self-renew?
- Figure 6h and i: Assume 2 examples of wt and Taf4a sKO regenerated muscle with each stain are shown? Need to have some indication/explanation of the frequency of the more mild or severe outcomes in Taf4a sKO as both are illustrated here.
- Supplementary Figure 1c: '(first row), in MuSCs on freshly isolated myofiber (second row), and in MuSCs after culturing myofibers for 3 days (third and fourth panel)' – should be columns or better way to indicate appropriate images. No statement about n= myofibres/mice examined. "8-20-weeks-old males and females were used, 4-6 weeks after tamoxifen administration" – not relevant?
- Supplementary Figure 2b: indicate the gene being measured on the panel.
- Supplementary Figure 6c: A statement is made in the text that EC cell proliferation was not decreased by knockdown of Taf4a, yet it is only n=2 and so no statistical testing is applied. Ideally, another repeat is performed or failing that, the text amended accordingly.
- Supplementary Figure 7: Although obvious to some, please clearly start which are the positive and negative antibody selections used for MuSC.

Version 1:

Reviewer comments:

Reviewer #1

(Remarks to the Author)

The authors' revision has improved the manuscript.

Given that TAF4a localizes at promoter regions, it is very surprising that ATAC-seq of Taf4asKO MuSCs displayed chromatin changes mainly located at intergenic and intronic regions. These observations suggest that modified chromatin accessibility in Taf4asKO MuSCs is an indirect effect. This point should be acknowledged and discussed.

Reviewer #2

(Remarks to the Author)

The authors have provided comprehensive experimental evidence addressing all the queries. Great piece of work. Congratulations!

REVIEWER COMMENTS

Reviewer #1 (Remarks to the Author):

Georgieva et al. report that TAF4A is critical for genome stability and quiescence of skeletal muscle stem cells. The findings are of interest and the experiments well-executed and controlled. However, the following points should be addressed before publication.

Response: We thank the reviewer for the positive evaluation of the manuscript and for the helpful comments.

Specific comments:

1. To avoid unintended MuSC activation, staining of uninjured control and TAF4asKO muscle sections should be performed to evaluate MyoD protein accumulation.

Response: Thank you for the comment. We provided this information in (Figure 1e) of the initial submission of the manuscript, which shows MyoD protein levels in uninjured control and *Taf4a*^{SKO} muscle sections.

2. ATAC-seq would determine whether heterochromatin reduction occurs at specific genomic regions and whether this phenomenon is associated with modified gene expression.

Response: To address the reviewer's question, we performed ATAC-seq of freshly isolated MuSCs from control and *Taf4a*^{SKO}. As expected, the coverage heat map revealed a more open chromatin organization in *Taf4a*^{SKO} MuSCs (new Supplemental Figure 3d). Bioinformatics analysis identified 5330 upregulated and 4068 downregulated ATAC-seq peaks in *Taf4a*^{SKO} compared to control MuSCs. However, only 13–15% of deregulated peaks were annotated to genes, indicating that changes are primarily occurring in repetitive and intergenic regions and to a lesser degree in regions that might be directly relevant for regulation of gene expression (Supplemental Figure 3e). Further analysis of the genomic distribution of peaks related to genes showed that only 3% are located in proximal promoter regions, whereas 51% are within intronic regions and 38% within intergenic regions (revised Supplemental Figure 3f). To determine whether differential ATAC-seq peaks correlate with modified gene expression, we overlapped differential ATAC-seq peaks with differentially expressed genes from our RNA-seq data. We detected a weak correlation between transcriptional changes and differential ATAC-seq peaks. Only 6.5% of transcriptionally altered genes associated with changes in DNA accessibility (revised Supplemental Figure 3g, h). Taken together, the comparative ATAC-seq analysis suggests that changes in chromatin accessibility after inactivation of *Taf4a* primarily occur in intronic and intergenic regions, most likely due to untimely activation of MuSCs and loss of H3K9me3, rather than being directly linked to changes in gene expression. The new findings are discussed in the revised text and shown in (revised Supplemental Figure 3).

3. Decreased H3K9me3 is expected to favor gene activation. Was H3K9me3 reduced at the epigenetic modifiers downregulated in TAF4asKO MuSCs? Is reduced H3K9me3 the result of indirect effects of TAF4A deletion?

Response: The reviewer raises an interesting question. To address this, we performed H3K9me3 CUT&RUN of both control and *Taf4a*^{SKO} MuSCs. We observed broad H3K9me3 peaks, sometimes spanning several megabases, which were notably reduced after inactivation of *Taf4a* (Figure 1a, b for reviewers). The size of these peaks is consistent with previously published data [1, 2]. However, we did not detect H3K9me3 peaks at transcription start sites (TSSs), including those of downregulated epigenetic modifiers (Figure 1c for reviewers). While we observed H3K4me3 peaks at the promoters of these genes, there was no corresponding enrichment of H3K9me3 (Figure 1d, e for reviewers). H3K9me3 is a hallmark of constitutive heterochromatin, typically associated with repetitive genomic

elements and structural regions rather than gene promoters. Therefore, the global reduction in H3K9me3 observed in *Taf4a*^{SKO} MuSCs likely reflects broader changes in chromatin structure and may contribute to reduced genome stability, rather than directly influencing the expression of specific epigenetic modifiers. Moreover, neither TAF4A nor KANSL2 directly binds to the *Suv39h2* locus, the gene encoding the H3K9 trimethyltransferase, supporting the conclusion that the observed changes in H3K9me3 are indirect consequences of *Taf4a* deletion (Figure 4 for reviewers). Taken together, our data suggest that reduced H3K9me3 in *Taf4a*^{SKO} MuSCs is not a direct regulatory mechanism for the downregulation of epigenetic modifiers, but instead reflects broader, possibly structural, chromatin changes.

Figure 1 for reviewers: **a, b** H3K9me3 distribution in control and *Taf4a*^{skO} MuSCs, showing broad peaks, characteristic for constitutive heterochromatin. A 28-Mb segment of the X chromosome is shown. **c** Heat maps of CUT&RUN signals for H3K9me3 at transcriptional start sites. The blue-to-red gradient indicates high-to-low counts in the corresponding region. **d, e** H3K9me3 and H3K4me3 distribution in the proximal promoter region of the *Smyd5*(d) and *Smarca4* (e) gene in control and *Taf4a*^{skO} MuSCs.

4. *Figure 3f: In addition to overexpressed, endogenous TAF4A-NFYB interaction should be documented.*

Response: Of course, the reviewer is right. Overexpression may create artifacts. To investigate whether endogenous TAF4A interacts with NF-YB, we performed immunoprecipitations in both C2C12 cells and primary myoblasts without any overexpression. We detected clear interactions between endogenous TAF4A and NF-YB/NF-YA, both in C2C12 cells and primary myoblasts, validating the potential physiological relevance of this interaction. We now show the data pertaining to the interactions of endogenous proteins and deleted the overexpression results (revised Figure 3f; revised Supplemental Figure 4b).

5. *Were the NF-YA occupied genes active in TAF4aKO MuSCs? If they were, what is the functional significance of NF-YA-TAF4A interaction at the approximately 700 genes (Figure 3g) bound by both NF-YA and TAF4A?*

Response: We thank the reviewer for the thoughtful question. Gene regulation by NF-YA certainly does not always depend on TAF4A, since the majority of NF-YA binding site does not show an interaction with TAF4A. We demonstrate in (Fig. 3h) that NF-YA generated 4,128 gene-annotated CUT&RUN peaks. TAF4A only generated 808 gene-annotated CUT&RUN peaks, of which 625 overlapped with NF-YA peaks. Only 686 out of 4,128 genes binding NF-YA showed differential expression after inactivation of *Taf4a*, whereas the vast majority (3442 genes), was not affected. Absence of differential gene expression does not necessarily mean that the respective genes are expressed in MuSCs. Thus, we analyzed whether the 3442 genes, which are bound by NF-YA but are not affected by the absence of TAF4A, are expressed in MuSCs. 1479 out of the 3442 genes had a TPM >0,5 and are therefore robustly expressed in both control and *Taf4a* mutant MuSCs (Figure 2a for reviewers). Comparative GO analysis revealed that genes bound by both NF-YA and TAF4A are enriched in biological processes such as translation, chromatin organization, apoptotic process, gene expression and transcription (Figure 2b for reviewers), whereas genes bound by NF-YA but not TAF4A are primarily associated with transcriptional regulation (Figure 2c for reviewers). Obviously, genes bound by NF-YA, regardless of whether TAF4A is bound, are predominantly involved in transcriptional regulation (Figure 2b, c for reviewers). Since expression of genes only bound by NF-YA but not TAF4A remains largely unchanged upon *Taf4a* inactivation, we concluded that the cooperation of NF-YA with TAF4A allows regulation of a small set of genes involved in transcription, foremost *Kansl2*, which is an integrative subunit of chromatin-modifying NSL complex. In contrast, other cooperation partners or mechanisms are employed for controlling genes that are bound by NF-YA but not TAF4A.

Figure 2 for reviewers: **a** Venn diagram of genes expressed in control (TPM>0.5) and *Taf4a* mutant MuSCs and of NF-YA-annotated CUT&RUN peaks that were not deregulated in *Taf4a^{skO}* MuSCs. **b** GO term analysis of genes only bound by NF-YA and expressed in MuSCs, based on P-values using EnrichR. **c** GO term analysis of genes bound only by NF-YA and are transcriptionally expressed in MuSCs, based on P-values using EnrichR.

6. H3K9me3 heterochromatin mediates nuclear stiffness (Nava et al. Cell 2020, <https://doi.org/10.1016/j.cell.2020.03.052>). To evaluate whether stiffness reduction of nuclei in TAF4asKO MuSCs is due to a direct consequence of reduced KANSL2 rather than indirect effects, TAF4asKO MuSCs may be reconstituted by expressing exogenous KANSL2 followed by AFM measurements.

Response: Following the reviewer's request, we overexpressed KANSL2 in *Taf4a^{skO}* MuSCs and performed AFM measurements. Exogenous expression of KANSL2 attenuated reduction of nuclear stiffness in *Taf4a*-deficient MuSCs (revised Figure 5j), confirming the direct role of KANSL2. The results are also consistent with findings presented in (Figure 5i), where KANSL2 overexpression attenuated nuclear blebbing and micronuclei formation in *Taf4a^{skO}* MuSCs. In addition, knockdown of *Kansl2* in wild type MuSCs phenocopied the effects observed after inactivation of *Taf4a* (Figure 5h). Together, these results demonstrate that reduction of nuclear stiffness and associated nuclear abnormalities following *Taf4a* inactivation are direct consequences of reduced expression of *Kansl2*, a key component of the NSL complex.

Reviewer #2 (Remarks to the Author):

The study by Georgieva et al., titled “Regulation of the NSL complex by TAF4 is critical for genome stability and quiescence of skeletal muscle stem cells” investigated the role of TATA-binding protein-associated factor, TAF4A, in the maintenance of heterochromatin and quiescent state of muscle stem cells (MuSCs). The authors demonstrated that TAF4A together with NF-YA regulate the expression of *Kansl2*. They further showed that KANSL2 as a component of the NSL complex, regulates the expression of key epigenetic modifiers in MuSCs. Using CUT&Run the authors also showed the enrichment of TAF4A and KANSL2 at various genes including those involved in cell cycle control and DNA replication, p53 pathway and apoptosis. Loss of TAF4A results in perturbed TAF4A-NFY-NSL axis, leading to genomic instability with increased gH2A.X protein and foci, and nuclear blebbing. The authors further demonstrated that loss of *Taf4a* in MuSCs causes atypical activation of MuSCs with impaired proliferation and increased apoptosis, ultimately leading to compromised muscle regeneration. Overall, the study has demonstrated the significance of heterochromatin and genomic stability via TAF4A-NFY-NSL epigenetic axis during MuSC maintenance. However, at places the authors did not provide strong evidence, and the interpretations are less substantiated. The manuscript could benefit from revisions on the following points.

Response: We thank the reviewer for the positive evaluation of the manuscript and the helpful comments.

Major Comments:

1. The manuscript has strongly emphasized the structural integrity of Lamin A/C, specifically by post-translational modifications. However, the authors did not present any evidence on change in the acetylation status of lamins in the absence of TAF4A, except the phosphorylation. The authors must show the differential solubility of lamins (refer to PMID: 31576060) in *Taf4a* KO and *Kansl2* knockdown or can immunoprecipitate with acetyl-lysine and probe with lamin.

Response: We thank the reviewer for addressing this important point. We fully agree that direct evidence of reduced lamin A/C acetylation is critical to support our conclusions. We were struggling for quite some time to overcome technical challenges, including identification of high-quality antibodies and the limited availability of material due to the severe phenotype of *Taf4a*-deficient MuSCs (reduced proliferation and increased apoptosis), which interferes with expansion of *Taf4a*^{SKO} MuSCs. Eventually, we successfully performed immunoprecipitation of lamin A/C, followed by probing with an anti-acetyl-lysine antibody. After normalization to total lamin A/C levels, we detected a clear reduction of lamin A/C acetylation in *Taf4a*-deficient MuSCs. We also observed a similar reduction in *Kansl2* knockdown cells, supporting the role of TAF4A and KANSL2 in regulating lamin A/C acetylation. The new data is shown in (revised Figure 5d; revised Supplemental Figure 5c).

2. In figure 2, comparative RNA-seq data, it is important to mention at what time point the activated and proliferating WT MuSCs were collected for comparing expression profiles with freshly isolated *Taf4a* KO. The authors also need to provide an analysis of the 428 genes that are differentially downregulated in *Taf4a* KO but upregulated in aMuSCs to understand and explain the phenotype of *Taf4a* KO MuSCs. Similarly, the authors need to provide a separate analysis, including GO dot plot (to understand the percent genes associated with a term) where they compared freshly isolated WT MuSCs with freshly isolated *Taf4a* KO MuSCs.

Response: Thank you for pointing out these shortcomings. The activated and proliferating wild type MuSCs used for the comparative RNA-seq analysis were cultured for 3 days *in vitro* after isolation. To address the reviewer’s request, we conducted Gene Ontology (GO) enrichment analysis of the 428 genes that were downregulated in *Taf4a*^{SKO} MuSCs but upregulated in activated WT MuSCs using Enrichr to

ensure consistency with our previous analyses. GO terms enriched among these genes include G1 to S cell cycle control, mechanisms associated with pluripotency, p53 signaling and delta notch signaling pathway. The result is included in (revised Figure 2b). The GO analysis in the original submission (Supplemental Figure 3b), which compared freshly isolated wild type and *Taf4a*^{SKO} MuSCs was performed using DAVID. To maintain consistency, we re-analyzed this dataset using Enrichr and provide the updated results in (revised Supplemental Figure 3b). The exact number of genes associated with each GO term is indicated in square brackets on the graphs for clarity. In both GO analyses, genes downregulated in *Taf4a*^{SKO} MuSCs were predominantly associated with cell cycle, DNA replication, and p53 signaling pathway, consistent with the observed phenotype of impaired proliferation and increased apoptosis. Importantly, subsequent CUT&RUN analysis identified direct targets of TAF4A, revealing that the majority of the effects are secondary to severe genomic instability, although some deregulated genes are directly bound by TAF4A.

3. *The authors showed the interaction of NF-YB with TAF4A and then went on to show the CUT&Run experiments with NF-YA. The authors should discuss about NF-YA/TAF4A interaction, Also, the interaction is shown in HEK293T, which is less significant due to the absence of any nuclear phenotype in embryonic fibroblasts. The authors must show the interaction of NF-YA/NF-YB with TAF4A, in either MuSCs or C2C12 myoblasts.*

Response: We thank the reviewer for this thoughtful and constructive comment. As described in the initial submission, NF-Y is a ubiquitously expressed heterotrimeric transcription factor (TF) complex composed of NF-YA, NF-YB, and NF-YC subunits. NF-YA contains both DNA-binding and transactivation domains, whereas NF-YB and NF-YC possess additional histone-fold domains that are structurally similar to corresponding domains within TAF4A (Supplemental Figure 4b). In our CUT&RUN experiments, de novo motif analysis of TAF4A-bound regions revealed strong enrichment of the NF-YA consensus motif, consistent with NF-YA's role in DNA recognition and recruitment of transcriptional regulators to chromatin. Our initial co-immunoprecipitation experiments in HEK293T cells demonstrated a robust interaction between NF-YB and TAF4A, which we interpret as likely mediated through the shared histone-fold domains. This is supported by prior studies showing that TAF4A forms complexes with other histone-fold domain-containing TAFs via "handshake" interactions [3]. We fully agree that validation of the interaction of NY-Y with TAF4A in muscle cells is essential. To address this issue, we performed co-immunoprecipitations of endogenous TAF4A in both C2C12 myoblasts and primary MuSCs. We detected clear interactions between endogenous TAF4A and NF-YB/NF-YA, both in C2C12 cells and primary myoblasts, validating the potential physiological relevance of this interaction. We now show the data pertaining to the interactions of endogenous proteins and deleted the overexpression results (revised Figure 3f; revised Supplemental Figure 4b).

4. *Heatmaps and genomic distribution of NF-YA need to be provided. Among the 48 NF-YA peaks lost in Taf4aSKO MuSCs, how many of these are differentially expressed in Taf4aSKO MuSCs.*

Response: We thank the reviewer for this comment. We now provide a heatmap and genome-wide distribution of NF-YA CUT&RUN peaks in wild-type MuSCs. The genomic distribution indicates that 34% of NF-YA binding occurs at promoters and 34% at intronic regions, which is in line with published findings, indicating that NF-YA localizes at both promoters and enhancers[4]. These data are included in (revised Supplemental Figure 4d, e). As requested, we also analyzed NF-YA CUT&RUN data from *Taf4a*^{SKO} MuSCs, which showed increased background signals, including higher IgG read counts, most likely due to the globally more open chromatin structure of mutant cells (Figure 3a for Reviewer). Nevertheless, we identified 333 statistically significant differential NF-YA peaks, of which 48 gene-annotated peaks were lost and 71 gained in *Taf4a* mutants. Differential peaks in *Taf4a*^{SKO} cells were enriched at 3'UTRs (45%) and 5'UTRs (23%), suggesting that many of these changes are unspecific and

unlikely to reflect canonical NF-YA functions (Figure 3b for Reviewers). Finally, we found that only 3 genes were differentially expressed when overlapping the 48 gene-annotated NF-YA peaks lost in *Taf4a*^{SKO} MuSCs with down-DEG in RNA-seq data, indicating that the majority of these binding events are not directly relevant for transcriptional regulation (Figure 3c for Reviewers).

Figure 3 for reviewers: **a** Heat maps of centered NF-YA CUT&RUN peaks in control and *Taf4a*^{SKO} MuSCs. The blue-to-red gradient indicates high-to-low counts in the corresponding region. **b** Genome-wide distribution of differential NF-YA peaks in *Taf4a*^{SKO} MuSCs. **c** Venn diagram of overlapping NF-YA lost CUT&RUN gene-annotated peaks in *Taf4a*^{SKO} MuSCs and RNA-seq down-DEG in *Taf4a*^{SKO} MuSCs.

5. In figure 2, the authors reported that among the 21 genes including *Smyd5* and *Suv39H2*, are potential direct targets of TAF4A. However, in Figure 4i, the authors mention about *Kansl2* as a regulator of these epigenetic modifiers. Surprisingly *Smyd5* and *Suv39H2* are missing in this gene list. It is important to dissect which of these are TAF4A and *Kansl2* targets. The authors can show that peak distribution for TAF4A and *Kansl2* at these genes.

Response: Thank you for the comment, which helped us to present our findings in a better way. We would like to clarify that we did not claim that KANSL2 directly regulates *Smyd5* and *Suv39h2* expression. We found that among the 21 epigenetic modifiers downregulated in *Taf4a* knockout cells, 14 are directly bound by KANSL2, but *Smyd5* and *Suv39h2* were not among them. IGV snapshots (Figure 4 for Reviewers) showed no binding of TAF4A or KANSL2 to promoter regions of *Smyd5* and *Suv39h2*, suggesting their downregulation is likely an indirect effect of the loss of *Taf4a*. We have improved the wording to avoid any misunderstanding.

Figure 4 for reviewers: a, b TAF4A, KANSL2 and H3K4me3 distribution in the proximal promoter region of the *Smyd5*(a) and *Suv39h2* (b) genes in WT MuSCs.

Minor Comments:

1. In the figure 1e, the *MyoD1* staining is less convincing due to non-specific background. The authors should use other antibodies to present this data.

Response: We agree with the reviewer that the MYOD1 staining in the initial (Figure 1e) shows some unspecific background, although the correct MYOD1 signal clearly co-localizes with DAPI-positive nuclei. We have replaced the incriminated images with more representative examples. We would also like to point out that we provided additional evidence for aberrant activation of MuSC after inactivation of *Taf4a*, i.e. immunofluorescent staining for PAX7 and the quiescence marker CALCR (Figure 1f). Furthermore, our RNA-seq data confirmed aberrant activation MuSC by demonstrating downregulation of quiescence-associated genes.

2. In figure 1B and 1C, the authors have a clearly shown a decrease in H3K9me3. Redundantly, the authors showed the same decrease in K9 trimethylation in the figure 2e and 2f. Though the authors were trying to corroborate that the decrease in methylation is due to *Smyd5* and *Suv39h2*, it is less informative. Immunofluorescence panels are missing scales.

Response: We understand the reviewer's concern. However, we wanted to demonstrate that inactivation of *Taf4a* does not only reduce H3K9 trimethylation in steady-state conditions (Figure 1) but also after induction of proliferation (Figure 2), which we consider as non-redundant information. In our view, this distinction is important, as it supports the conclusion that the reduction of H3K9me3 depends on TAF4A and not on changes in cell state. We apologize for the oversight regarding scale bars for the immunofluorescence images. Indeed, only the last panel had a scale bar. We have now added the missing scale bars to all relevant images in the revised manuscript.

3. It will be informative to show all the GO plots in a dot plot to understand the number of genes associated with a term.

Response: We thank the reviewer for the suggestion. To maintain consistency across our analyses, we have chosen the EnrichR platform, which also provides comprehensive enrichment statistics, including the exact number of genes associated with each GO term. We have now included the number of genes associated with each term directly on the GO plots. This information is provided in square brackets next

to each GO term, easily allowing readers to assess the number of gens related to individual GO terms, while maintaining consistency with the original format of analysis.

4. In figure 6a is not truly representing the plotted graph. 3-days post isolation, there are very few MuSCs on the fiber. Similarly, figure 6d is not truly representing the graph. The authors should consider providing representative images.

Response: We are sorry for the confusion, which resulted from an error in labeling the y-axis in (Figure 6a). The graph represents the percentage of Pax7⁺/EdU⁺ cells found on 50 fibers, not the number of cells per fiber. The reference to "on 50 myofibers" was unintentionally omitted from the axis label. We have corrected this error in the revised version to clarify the basis for the quantification. Only very few MuSCs can be found on fibers of *Taf4a* knockout mice, since inactivation of *Taf4a* compromises proliferation and increases apoptosis, leading to reduction of Pax7⁺/EdU⁺ cells. We agree with the reviewer that the image in (Figure 6d) does not closely match the quantification. We have replaced the respective image, now showing an example that aligns with the quantification.

Reviewer #3 (Remarks to the Author):

A very interesting study demonstrating that TAF4A-NF-Y safeguards heterochromatin and genome stability of skeletal muscle satellite cells via the NSL complex. This has relevance not only for basic muscle biology but also for TAF4-related neuro-developmental disorders in human, which have not been reported to have direct muscular involvement. I recommend publication in Nature Communications.

Response: We thank the reviewer for the positive evaluation of the manuscript and for the helpful comments.

Major Comment

• *Other evidence is supportive (lamin phosphorylation changes, nuclear stiffness and structural changes) but is it possible to also directly show altered acetylation of lamin as a consequence of Taf4a knockout to further support the mechanism?*

Response:

The reviewer raises the same important point as reviewer #2 (comment #1). We fully agree that direct evidence of reduced lamin A/C acetylation is critical to support our conclusions. We were struggling for quite some time to overcome several technical challenges, including the difficulty in finding high-quality antibodies and the limited availability of material due to the severe phenotype of Taf4a-deficient MuSCs (reduced proliferation and increased apoptosis), which interferes with expansion of *Taf4a*^{SKO} MuSCs. Eventually, we successfully performed immunoprecipitation of lamin A/C, followed by probing with an anti-acetyl-lysine antibody. After normalization to total lamin A/C levels, we detected a clear reduction of lamin A/C acetylation in Taf4a-deficient MuSCs. We also observed a similar reduction in *Kansl2* knockdown cells, supporting the role of TAF4A and KANSL2 in regulating lamin A/C acetylation. The new data is shown in (revised Figure 5d; revised Supplemental Figure 5c).

Other Comments

• *Might be useful to state the non-specific lethal (NSL) complex on first mention in the abstract.*

Response: Thanks for the suggestion. We have revised the abstract to define the non-specific lethal (NSL) complex when first mentioned.

• *Would include ‘satellite cell’ when defining skeletal muscle stem cells (MuSCs) for clarity as several other cell types have been proposed as muscle stem cells over the years. Also, ‘satellite cell’ is only mentioned in the material and methods otherwise.*

Response: True, thanks for the remark. We now mention the term ‘satellite cell’ when first defining skeletal muscle stem cells (MuSCs).

• *Graphs in general have enough room to place X axis labels of ‘Control’ and ‘TAF4a SKO’ under appropriate column – TAF4a SKO text in red rather than dots?*

Response: Thank you for the suggestion. We have revised the graphs and place the X-axis labels ‘Control’ and ‘TAF4a^{SKO}’ directly under the corresponding columns.

- *Could comment on how well the NSL complex is conserved in human.*

Response: We have added a comment that the NSL complex is highly conserved in humans compared to mice. Key components and functions are essentially identical across species, supporting its fundamental role in gene regulation.

- *No description in material and methods of how muscle fibres were isolated.*

Response: We are sorry for this negligence. We have amended the Materials and Methods section to include a detailed description of the muscle fiber isolation protocol and culturing conditions.

- *If possible, including a model would be useful (graphical abstract format).*

Response: Thank you for the suggestion. Depiction of a model is certainly helpful. We have included a graphical model summarizing our findings in (new Figure 7).

- *Figure 1: Here and elsewhere, could n=3 be explained: assume it is n=3 mice but how many satellite cells were examined in Fig. 1a per mouse?*

Response: The reviewer is correct when assuming that “n=” **always** refers to the number of mice from cells were derived. We have now added the number of satellite cells analyzed per mouse to the figure legend of (revised Figure 1a) for clarity.

- *Supplementary Figure 1c: TAF4a immunolabeling in wt PAX7-containing satellite cells should instead be in Figure 1 as data presented is fundamental to the paper. Would suggest a higher power image of a satellite cell on a muscle fibre with surrounding myonuclei to better show the higher TAF4a protein levels in the satellite cell.*

Response: Thank you for the suggestion. We agree that it would be helpful to show an enlarged staining for TAF4a in wild-type PAX7-positive satellite cells in the main (revised Figure 1). However, (revised Figure 1) is already pretty crowded with 7 different panels. Including an additional panel would worsen the situation. We hope the reviewer agrees.

- *Figure 1c: bands look a bit strange e.g. H3K9me3 in Taf4a sKO. Here and elsewhere (e.g. Fig 2f, 4e, 5c,d, 6f), where just bands excised from a gel are presented, the whole gel should be shown in each case as Supplementary data.*

Response: Thank you for the comment. We appreciate the reviewer’s concern regarding presentation of the western blot bands in (Figure 1c) and other panels (e.g., Figures 2f; 4e; 5c, d; 6f). Of course, we will follow the journal's guidelines and provide full, uncropped images of all western blots in the source data file, ensuring transparency and reproducibility.

- *Figure 1f: calcitonin receptor (CalcR) expression not only labels quiescent satellite cells (e.g. doi:10.1371/journal.pone.0005205).*

Response: Thank you for the comment. While the study referenced by the reviewer (doi:10.1371/journal.pone.0005205) initially suggested that Calcitonin Receptor (Calcr) labels quiescent and activated muscle stem cells, it is an old study conducted exclusively *ex vivo* on cultured myofibers, with notable limitations. Specifically, it lacks quantification, shows background staining in

other myonuclei, and does not clearly describe the method for isolating satellite cells from cultured myofibers or assessing the purity of quiescent and activated MuSCs used for RT-PCR. In contrast, more recent and comprehensive studies have demonstrated that Calcr is exclusively expressed in quiescent MuSCs[5], and that its signaling contributes to the maintenance of quiescence in vivo[6]. Further mechanistic studies have confirmed the role of the Calcr–cAMP–PKA–Yap1 axis in sustaining quiescence[7]. Mutant mouse models have demonstrated the functional requirement of Calcr for MuSC quiescence [8]. We believe it is appropriate to refer to Calcr as a marker for quiescent MuSCs.

• *Figure 3h: would indicate in figure legend that CnR is CUT&Run.*

Response: Thank you. We updated the figure legend for (Figure 3h) to indicate that CnR refers to CUT&RUN.

• *Figure 4b: why are changes in others factors so inconsistent?*

Response: Thank you for the question. The seemingly inconsistencies in Figure 4b result from the nature of heatmap visualizations. Heatmaps use color to represent values, and small variations, especially when log-scaled, can appear more pronounced than they are. This can sometimes give the impression of inconsistency, even when the underlying quantitative changes are relatively minor.

• *Figure 6: using the isolated myofibre assay, it would be informative to know the fate of the associated satellite cells in the Taf4a sKO after 3 days. Is there change in the ratio of those that differentiate versus self-renew?*

Response: Thank you for the comment. We described that inactivation of Taf4a in MuSC leads to massive genome instability, eventually leading to apoptosis. *Taf4a*^{sKO} MuSCs are unable to either differentiate or self-renew, they do not survive. The inability of *Taf4a*^{sKO} MuSCs to differentiate or self-renew is reflected by the complete failure of muscle regeneration in *Taf4a*^{sKO} mice.

• *Figure 6h and i: Assume 2 examples of wt and Taf4a sKO regenerated muscle with each stain are shown? Need to have some indication/explanation of the frequency of the more mild or severe outcomes in Taf4a sKO as both are illustrated here.*

Response: We apologize for the confusion, which is due to incomplete labeling. The first set images (revised Figure 6h) depicts uninjured and injured TA muscles from control and *Taf4a*^{sKO} mice after H&E staining, whereas the second set of images (revised Figure 6i) shows uninjured and injured TA muscles from control and *Taf4a*^{sKO} mice after trichrome staining. We have improved the labelling to clearly distinguish between uninjured and CTX-injured conditions. All outcomes in *Taf4a*^{sKO} mice were severe: there is no variation in phenotype severity. Muscle regeneration is consistently and profoundly impaired in the absence of *Taf4a* across all analyzed samples.

• *Supplementary Figure 1c: '(first row), in MuSCs on freshly isolated myofiber (second row), and in MuSCs after culturing myofibers for 3 days (third and fourth panel)' – should be columns or better way to indicate appropriate images. No statement about n= myofibres/mice examined. "8-20-weeks-old males and females were used, 4-6 weeks after tamoxifen administration" – not relevant?*

Response: Thank you for the careful reading. We have revised the figure legend and now include the n-numbers for both myofibers and mice. The reviewer is also right regarding the statement about “8–20-

week-old males and females, 4–6 weeks after tamoxifen administration”—this was mistakenly left in figure during reorganization and not relevant for this figure. Accordingly, we have removed this information.

- *Supplementary Figure 2b: indicate the gene being measured on the panel.*

Response: Thanks again for the careful reading. Should have been our job We have added the name of the gene that was analyzed by RT-qPCR in (revised Supplemental Figure 2b).

- *Supplementary Figure 6c: A statement is made in the text that EC cell proliferation was not decreased by knockdown of Taf4a, yet it is only n=2 and so no statistical testing is applied. Ideally, another repeat is performed or failing that, the text amended accordingly.*

Response: Thank you for the comment. As noted, we analyzed two biological replicates in Supplementary Figure 6c. We did not observe an effect on mESC proliferation, although the small sample size did not permit a statistical evaluation. We have not performed an additional repeat since the result is consistent with two previously published studies reporting no effect of *Taf4a* loss on mESCs proliferation [9, 10]. We have amended the text as recommended to cope with the missing statistical evaluation.

- *Supplementary Figure 7: Although obvious to some, please clearly start which are the positive and negative antibody selections used for MuSC.*

Response: Thank you for the comment. We have updated the legend for (Supplemental Figure 7) to clearly indicate, which antibodies were used for positive and negative selection of MuSCs.

References

1. Becker, J.S., D. Nicetto, and K.S. Zaret, *H3K9me3-Dependent Heterochromatin: Barrier to Cell Fate Changes*. Trends Genet, 2016. **32**(1): p. 29-41.
2. Yokobayashi, S., et al., *Inherent genomic properties underlie the epigenomic heterogeneity of human induced pluripotent stem cells*. Cell Rep, 2021. **37**(5): p. 109909.
3. Xu, Y.L., et al., *A TFIID-SAGA Perturbation that Targets MYB and Suppresses Acute Myeloid Leukemia*. Cancer Cell, 2018. **33**(1): p. 13-+.
4. Oldfield, A.J., et al., *Histone-fold domain protein NF-Y promotes chromatin accessibility for cell type-specific master transcription factors*. Mol Cell, 2014. **55**(5): p. 708-22.
5. Yamaguchi, M., et al., *Calcitonin Receptor Signaling Inhibits Muscle Stem Cells from Escaping the Quiescent State and the Niche*. Cell Rep, 2015. **13**(2): p. 302-14.
6. Baghdadi, M.B., et al., *Reciprocal signalling by Notch-Collagen V-CALCR retains muscle stem cells in their niche*. Nature, 2018. **557**(7707): p. 714-+.
7. Zhang, L.D., et al., *The CalcR-PKA-Yap1 Axis Is Critical for Maintaining Quiescence in Muscle Stem Cells*. Cell Reports, 2019. **29**(8): p. 2154-+.
8. Zhang, L., et al., *Dlk1 regulates quiescence in calcitonin receptor-mutant muscle stem cells*. Stem Cells, 2021. **39**(3): p. 306-317.
9. Bahat, A., et al., *TAF4b and TAF4 differentially regulate mouse embryonic stem cells maintenance and proliferation*. Genes Cells, 2013. **18**(3): p. 225-37.
10. Langer, D., et al., *Essential role of the TFIID subunit TAF4 in murine embryogenesis and embryonic stem cell differentiation*. Nat Commun, 2016. **7**: p. 11063.

Responses to reviewers' comments

Reviewer #1 (Remarks to the Author):

The authors' revision has improved the manuscript.

Given that TAF4a localizes at promoter regions, it is very surprising that ATAC-seq of Taf4asKO MuSCs displayed chromatin changes mainly located at intergenic and intronic regions. These observations suggest that modified chromatin accessibility in Taf4asKO MuSCs is an indirect effect. This point should be acknowledged and discussed.

Response: We thank the reviewer for the positive feedback and the comment. We completely agree that changes in chromatin accessibility in *Taf4a*^{sKO} MuSCs are to a large extent due to secondary effects. Major parts of the study are dedicated to unravel the cause of such indirect effects that happen in part due to the diminished expression of *Kansl2*, reducing activity of the NSL complex. The NSL complex not only acetylates Lamin but also plays an important role in chromatin decompaction by generating H4K16ac. Furthermore, we assume that the untimely activation of MuSCs and the concomitant loss of H3K9me3 contributes. We already considered the indirect effects of *Taf4a* inactivation in the previous version of the manuscript but are happy to extend the discussion. We now explicitly address the indirect effects in the results section: “...suggesting that the altered chromatin accessibility in *Taf4a*^{sKO} MuSCs is most likely caused by indirect effects.”

In addition, we extend our previous statement in the discussion—“Numerous promoters of epigenetic modifiers (excluding *Smyd5* and *Suv39h2*), downregulated in *Taf4a*^{sKO} MuSCs, showed binding of *KANSL2*, suggesting that their attenuated expression is caused by reduced activity of the *Kansl2* gene, secondary to the loss of *Taf4a*”—by adding: “The ATAC-seq data revealed changes in chromatin accessibility predominantly at intergenic and intronic regions, which are most likely indirect consequences of transcriptional and epigenetic perturbations, rather than direct effects of *Taf4a* inactivation.”